# Non-Asymptotic Analysis for Two Time-scale TDC with General Smooth Function Approximation

**Yue Wang**
Department of Electrical Engineering
University at Buffalo
Buffalo, NY, USA
ywang294@buffalo.edu

**Shaofeng Zou**
Department of Electrical Engineering
University at Buffalo
Buffalo, NY, USA
szou3@buffalo.edu

**Yi Zhou**
Department of Electrical and Computer Engineering
University of Utah
Salt Lake City, Utah, USA
yi.zhou@utah.edu

## Abstract

Temporal-difference learning with gradient correction (TDC) is a two time-scale algorithm for policy evaluation in reinforcement learning. This algorithm was initially proposed with linear function approximation, and was later extended to the one with general smooth function approximation. The asymptotic convergence for the on-policy setting with general smooth function approximation was established in [Bhatnagar et al., 2009], however, the non-asymptotic convergence analysis remains unsolved due to challenges in the non-linear and two-time-scale update structure, non-convex objective function and the projection onto a time-varying tangent plane. In this paper, we develop novel techniques to address the above challenges and explicitly characterize the non-asymptotic error bound for the general off-policy setting with i.i.d. or Markovian samples, and show that it converges as fast as $\mathcal{O}(1/\sqrt{T})$ (up to a factor of $\mathcal{O}(\log T)$). Our approach can be applied to a wide range of value-based reinforcement learning algorithms with general smooth function approximation.

## 1 Introduction

In reinforcement learning (RL), an agent interacts with a stochastic environment in order to maximize the total reward [Sutton and Barto, 2018]. Towards this goal, it is often needed to evaluate how good a policy performs, and more specifically, to learn its value function. Temporal difference (TD) learning algorithm is one of the most popular policy evaluation approaches. However, when applied with function approximation approach and/or under the off-policy setting, the TD learning algorithm may diverge [Baird, 1995, Tsitsiklis and Van Roy, 1997]. To address this issue, a family of gradient-based TD (GTD) algorithms, e.g., GTD, GTD2, temporal-difference learning with gradient correction (TDC) and Greedy-GQ, were developed for the case with linear function approximation [Maei, 2011, Sutton et al., 2009b, Maei et al., 2010, Sutton et al., 2009a,b]. These algorithms were later extended to the case with general smooth function approximation in [Bhatnagar et al., 2009], where asymptotic convergence guarantee was established for the on-policy setting with i.i.d. samples.

Despite the success of the GTD methods in practice, previous theoretical studies only showed that these algorithms converge asymptotically, and did not suggest how fast these algorithms converge and how the accuracy of the solution depends on various parameters of the algorithms. Not until

35th Conference on Neural Information Processing Systems (NeurIPS 2021).

recently have the non-asymptotic error bounds for these algorithms been investigated, e.g., [Dalal et al., 2020, Karmakar and Bhatnagar, 2018, Wang and Zou, 2020, Xu et al., 2019, Kaledin et al., 2020, Dalal et al., 2018, Wang et al., 2017], which mainly focus on the case with linear function approximation. These results thus cannot be directly applied to more practical applications with general smooth function approximation, e.g., neural networks, which have greater representation power, do not need to construct feature mapping, and are widely used in practice.

In this paper, we develop a non-asymptotic analysis for the TDC algorithm with general smooth function approximation (which we refer to as non-linear TDC) for both i.i.d. and Markovian samples. Technically, the analysis in this paper is not a straightforward extension of previous studies on those GTD algorithms with linear function approximation. First of all, different from existing studies with linear function approximation whose objective functions are *convex* and the updates are *linear*, the objective function of the non-linear TDC algorithm is *non-convex*, and the two time-scale updates are *non-linear* functions of the parameters. Second, the objective function of the non-linear TDC algorithm, the mean-square projected Bellman error (MSPBE), involves a projection onto a *time-varying* tangent plane which depends on the sample trajectory, whereas for GTD algorithms with linear function approximation, this projection is time-invariant. Third, due to the two time-scale structure of the algorithm and the Markovian noise, novel techniques to deal with the stochastic bias and the tracking error need to be developed.

## 1.1 Challenges and Contributions

In this section, we summarize the technical challenges and our contributions.

**Analysis for two time-scale non-linear updates and non-convex objective.** Unlike many existing results on two time-scale stochastic approximation, e.g., [Konda et al., 2004, Gupta et al., 2019, Kaledin et al., 2020] and the studies of linear GTD algorithms in [Xu et al., 2019, Ma et al., 2020, Wang et al., 2017, Dalal et al., 2020], the objective function of the non-linear TDC is non-convex, and its two time-scale updates are non-linear. Therefore, existing studies on linear two time-scale algorithms cannot be directly applied. Moreover, the convergence to global optimum cannot be guaranteed for the non-linear TDC algorithm, and therefore, we study the convergence to stationary points. In this paper, we develop a novel non-asymptotic analysis of the non-linear TDC algorithm, which solves RL problems from a non-convex optimization perspective. We note that our analysis is not a straightforward extension of analyses of non-convex optimization, as the update rule here is two time-scale and the noise is Markovian. The framework we develop in this paper can be applied to analyze a wide range of value-based RL algorithms with general smooth function approximation.

**Time-varying projection.** For the MSPBE, a projection of the Bellman error onto the parameterized function class is involved. However, unlike linear function approximation, the projection onto a general smooth class of functions usually does not have a closed-form solution. Thus, a projection onto the tangent plane at the current parameter is used instead, which incurs a time-varying projection that depends on the current parameter and thus the sample trajectory. This brings in additional challenges in the bias and variance analysis due to such dependency. We develop a novel approach to decouple such a dependency and characterize the bias by exploiting the uniform ergodicity of the underlying MDP and the smoothness of the parameterized function. The new challenges posed by the time-varying projection and the dependence between the projection and the sample trajectory are not special to the non-linear TDC investigated in this paper, and they exist in a wide range of value-based algorithms with general smooth function approximation, where our techniques can be applied.

**A tight tracking error analysis.** Due to the two time-scale structure of the update rule, the tracking error, which measures how fast the fast time-scale tracks its own limit, needs to be explicitly bounded. Unlike the studies on two time-scale linear stochastic approximation [Dalal et al., 2020, Kaledin et al., 2020, Konda et al., 2004], where a linear transformation can asymptotically decouple the dependence between the fast and slow time-scale updates, it is non-trivial to construct such a transformation for non-linear updates. To develop a tight bound on the tracking error, we develop a novel technique that bounds the tracking error as a function of the gradient of the MSPBE. This leads to a tighter bound on the tracking error compared to many existing works on two time-scale analysis, e.g., [Wu et al., 2020, Hong et al., 2020]. Although we do not decouple the fast and slow time-scale updates, we still obtain a desired convergence rate of $\mathcal{O}(1/\sqrt{T})$ (up to a factor of $\log T$), which matches with the complexity of stochastic gradient descent for non-convex problems [Ghadimi and Lan, 2013].

## 1.2 Related Work

**TD, Q-learning and SARSA.** The asymptotic convergence of TD with linear function approximation was shown in [Tsitsiklis and Van Roy, 1997], and the non-asymptotic analysis of TD was developed in [Srikant and Ying, 2019, Lakshminarayanan and Szepesvari, 2018, Bhandari et al., 2018, Dalal et al., 2018, Sun et al., 2020]. Moreover, [Cai et al., 2019] further studied the non-asymptotic error bound of TD learning with neural function approximation. Q-learning and SARSA are usually used for solving the optimal control problem and were shown to converge asymptotically under some conditions in [Melo et al., 2008, Perkins and Precup, 2003]. Their non-asymptotic error bounds were also studied in [Zou et al., 2019]. The non-asymptotic analysis of Q-learning under the neural function approximation was developed in [Cai et al., 2019, Xu and Gu, 2020]. Note that all these algorithms are one time-scale, while the TDC algorithm we study is a two time-scale algorithm.

**GTD methods with linear function approximation.** A class of GTD algorithms were proposed to address the divergence issue for off-policy training [Baird, 1995] and arbitrary smooth function approximation [Tsitsiklis and Van Roy, 1997], e.g., GTD, GTD2 and TDC [Maei, 2011, Sutton et al., 2009b, Maei et al., 2010, Sutton et al., 2009a,b]. Recent studies established their non-asymptotic convergence rate, e.g., [Dalal et al., 2018, Wang et al., 2017, Liu et al., 2015, Gupta et al., 2019, Xu et al., 2019, Dalal et al., 2020, Kaledin et al., 2020, Ma et al., 2020, Wang and Zou, 2020, Ma et al., 2021] under i.i.d. and Markovian settings. These studies focus on the case with linear function approximation, and thus the objective functions are convex, and the updates are linear. In this paper, we focus on the non-linear TDC algorithm with general smooth function approximation, where the two time-scale update rule is non-linear, the objective is non-convex, and the projection is time-varying, and thus new techniques are required to develop the non-asymptotic analysis.

**Non-linear two time-scale stochastic approximation.** There are also studies on asymptotic convergence rate and non-asymptotic analysis for non-linear two time-scale stochastic approximation, e.g., [Mokkadem et al., 2006, Doan, 2021]. Although the non-linear update rule is investigated, it is assumed that the algorithm converges to the global optimum. In this paper, we do not make such an assumption on the global convergence, which may not necessarily hold for the non-linear TDC algorithm, and instead, we study the convergence to stationary points, which is a widely used convergence criterion for non-convex optimization problems. We also note that there is a resent work studying the batch-based non-linear TDC in [Xu and Liang, 2021], where at each update, a batch of samples is used. To achieve a sample complexity of $\mathcal{O}(\epsilon^{-2})$, a batch size of $\mathcal{O}(\epsilon^{-1})$ is required in [Xu and Liang, 2021] to control the bias and variance. We note that by setting the batch size being one in [Xu and Liang, 2021], the desired sample complexity cannot be obtained, and their error bound will be a constant. In this paper, we focus on the non-linear TDC algorithm without using the batch method, where the parameters update in an online and incremental fashion and at each update only one sample is used. Our error analysis is novel and more refined as it does not require a large batch size of $\mathcal{O}(\epsilon^{-1})$ while still achieving the same sample complexity.

## 2 Preliminaries

### 2.1 Markov Decision Process

A Markov decision process (MDP) is a tuple $(\mathcal{S}, \mathcal{A}, \mathsf{P}, r, \gamma)$, where $\mathcal{S}$ and $\mathcal{A}$ are the state and action spaces, $\mathsf{P} = \mathsf{P}(s'|s, a)$ is the transition kernel, $r : \mathcal{S} \times \mathcal{A} \times \mathcal{S} \to \mathbb{R}^+$ is the reward function bounded by $r_{\max}$, and $\gamma \in [0, 1]$ is the discount factor. A stationary policy $\pi$ maps a state $s \in \mathcal{S}$ to a probability distribution $\pi(\cdot|s)$ over the action space $\mathcal{A}$. At each time-step $t$, suppose the process is at some state $s_t \in \mathcal{S}$, and an action $a_t \in \mathcal{A}$ is taken. Then the system transits to the next state $s_{t+1}$ following the transition kernel $\mathsf{P}(\cdot|s_t, a_t)$, and the agent receives a reward $r(s_t, a_t, s_{t+1})$.

For a given policy $\pi$ and any initial state $s \in \mathcal{S}$, we define its value function as $V^\pi(s) = \mathbb{E}\left[\sum_{t=0}^{\infty} \gamma^t r(S_t, A_t, S_{t+1})|S_0 = s, \pi\right]$. The goal of policy evaluation is to use the samples generated from the MDP to estimate the value function. The value function satisfies the Bellman equation: $V^\pi(s) = T^\pi V^\pi(s)$ for any $s \in \mathcal{S}$, where the Bellman operator $T^\pi$ is defined as

$$T^\pi V(s) = \sum_{s' \in \mathcal{S}, a \in \mathcal{A}} \mathsf{P}(s'|s, a)\pi(a|s)r(s, a, s') + \gamma \sum_{s' \in \mathcal{S}, a \in \mathcal{A}} \mathsf{P}(s'|s, a)\pi(a|s)V(s'). \quad (1)$$

Hence the value function $V^\pi$ is the fixed point of the Bellman operator $T^\pi$ [Bertsekas, 2011].

## 2.2 Function Approximation

In practice, the state space $\mathcal{S}$ usually contains a large number of states or is even continuous, which will induce a heavy computational overhead. A popular approach is to approximate the value function using a parameterized class of functions. Consider a parameterized family of functions $\left\{V_\theta : \mathcal{S} \to \mathbb{R} | \theta \in \mathbb{R}^N\right\}$, e.g., neural networks. The goal is to find a $V_\theta$ with a compact representation in $\theta$ to approximate the value function $V^\pi$. In this paper, we focus on a general family of smooth functions, which may not be linear in $\theta$.

## 3 TDC with Non-Linear Function Approximation

In this section, we introduce the TDC algorithm with general smooth function approximation in [Bhatnagar et al., 2009] for the off-policy setting with both i.i.d. samples and Markovian samples, and further characterize the non-asymptotic error bounds.

Consider the the following mean-square projected Bellman error (MSPBE):

$$J(\theta) = \mathbb{E}_{\mu^\pi}\left[\|V_\theta(s) - \mathbf{\Pi}_\theta T^\pi V_\theta(s)\|^2\right], \tag{2}$$

where $\mu^\pi$ is the stationary distribution induced by the policy $\pi$, and $\mathbf{\Pi}_\theta$ is the orthogonal projection onto the tangent plane of $V_\theta$ at $\theta$: $\left\{\hat{V}_\zeta(s) | \zeta \in \mathbb{R}^N \text{ and } \hat{V}_\zeta(s) = \phi_\theta(s)^\top \zeta\right\}$ and $\phi_\theta(s) = \nabla V_\theta(s)$. Note that the projection is onto the tangent plane instead of $\left\{V_\theta : \theta \in \mathbb{R}^N\right\}$ since the projection onto the latter one may not be computationally tractable if $V_\theta$ is non-linear.

In [Bhatnagar et al., 2009], the authors proposed a two time-scale TDC algorithm to minimize the MSPBE $J(\theta)$. Specifically, a stochastic gradient descent approach is used with the weight doubling trick (for the double sampling problem) [Sutton et al., 2009a], which yield a two time-scale update rule. We note that the algorithm developed in [Bhatnagar et al., 2009] was for the on-policy setting with i.i.d. samples from the stationary distribution, and the asymptotic convergence of the algorithm to stationary points was established.

In the off-policy setting, the goal is to estimate the value function $V^\pi$ of the target policy $\pi$ using the samples from a different behavior policy $\pi_b$. In this case, the MSPBE can be written as

$$J(\theta) = \mathbb{E}_{\mu^{\pi_b}}[\|V_\theta(s) - \mathbf{\Pi}_\theta T^\pi V_\theta(s)\|^2], \tag{3}$$

and we use the approach of importance sampling. Following steps similar to those in [Maei, 2011], $J(\theta)$ can be further written as

$$J(\theta) = \mathbb{E}_{\mu^{\pi_b}}[\rho(S, A)\delta_{S,A,S'}(\theta)\phi_\theta(S)]^\top A_\theta^{-1} \mathbb{E}_{\mu^{\pi_b}}[\rho(S, A)\delta_{S,A,S'}(\theta)\phi_\theta(S)], \tag{4}$$

where $\delta_{s,a,s'}(\theta) = r(s, a, s') + \gamma V_\theta(s') - V_\theta(s)$ is the TD error, $\phi_\theta(s) = \nabla V_\theta(s)$ is the character vector, $\rho(s, a) = \frac{\pi(a|s)}{\pi_b(a|s)}$ is the importance sampling ratio for a given sample $O = (s, a, r, s')$ and $A_\theta = \mathbb{E}_{\mu^{\pi_b}}[\phi_\theta(S)\phi_\theta(S)^\top]$.

To compute $\nabla J(\theta)$, we consider its $i$-th entry, i.e., the partial derivative w.r.t. the $i$-th entry of $\theta$:

$$-\frac{1}{2}\frac{\partial J(\theta)}{\partial \theta^i}$$
$$= \underbrace{-\mathbb{E}_{\mu^{\pi_b}}\left[\frac{\partial}{\partial \theta^i}(\rho\delta\phi)\right]^\top A_\theta^{-1}\mathbb{E}_{\mu^{\pi_b}}[\rho\delta\phi]}_{(a)} + \underbrace{\frac{1}{2}(A_\theta^{-1}\mathbb{E}_{\mu^{\pi_b}}[\rho\delta\phi])^\top \mathbb{E}_{\mu^{\pi_b}}\left[\frac{\partial}{\partial \theta^i}(\phi\phi^\top)\right](A_\theta^{-1}\mathbb{E}_{\mu^{\pi_b}}[\rho\delta\phi])}_{(b)},$$
$$\tag{5}$$

where to simplify notations, we omit the dependence on $\theta, S, A$ and $S'$. To get an unbiased estimate of the terms in (5), several independent samples are needed, but this is not applicable when there is only one sample trajectory. Hence we employ the weight doubling trick [Sutton et al., 2009a]. Define $\omega(\theta) = A_\theta^{-1}\mathbb{E}_{\mu^{\pi_b}}[\rho(S, A)\delta_{S,A,S'}(\theta)\phi_\theta(S)]$, then term $(a)$ can be written as follows:

$$-\mathbb{E}_{\mu^{\pi_b}}\left[\frac{\partial}{\partial \theta^i}(\rho\delta\phi)\right]^\top A_\theta^{-1}\mathbb{E}_{\mu^{\pi_b}}[\rho\delta\phi]$$

$$= -\mathbb{E}_{\mu^{\pi_b}}\left[\rho(\gamma(\phi_\theta(S'))_i - (\phi_\theta(S))_i)\phi_\theta(S)\right]^\top \omega(\theta) - \mathbb{E}_{\mu^{\pi_b}}\left[\rho\delta(\nabla^2 V)_i\right]^\top \omega(\theta); \qquad (6)$$

and term $(b)$ can be written as follows:

$$(A_\theta^{-1}\mathbb{E}_{\mu^{\pi_b}}[\rho\delta\phi])^\top \mathbb{E}_{\mu^{\pi_b}}\left[\frac{\partial}{\partial\theta^i}(\phi\phi^\top)\right](A_\theta^{-1}\mathbb{E}_{\mu^{\pi_b}}[\rho\delta\phi]) = 2\mathbb{E}_{\mu^{\pi_b}}\left[\phi^\top\omega(\theta)(\frac{\partial}{\partial\theta^i}\phi^\top)\omega(\theta)\right]. \quad (7)$$

Hence the gradient can be re-written as

$$-\frac{\nabla J(\theta)}{2} = \mathbb{E}_{\mu^{\pi_b}}\left[\rho(S,A)\delta_{S,A,S'}(\theta)\phi_\theta(S)\right] - h(\theta,\omega(\theta)) - \gamma\mathbb{E}_{\mu^{\pi_b}}\left[\rho(S,A)\phi_\theta(S')\phi_\theta(S)^\top\right]\omega(\theta),$$
$$(8)$$

where $h(\theta,\omega) = \mathbb{E}_{\mu^{\pi_b}}[(\rho(S,A)\delta_{S,A,S'}(\theta) - \phi_\theta(S)^\top\omega)\nabla^2 V_\theta(S)\omega]$. Thus with this weight doubling trick [Sutton et al., 2009a], a two time-scale stochastic gradient descent algorithm can be constructed. In Algorithm 1, we present the algorithm for the Markovian setting. The algorithm under the i.i.d. setting is slightly different, hence we refer the readers to Algorithm 2 in Appendix B.

---

**Algorithm 1** Non-Linear Off-Policy TDC under the Markovian Setting

---

**Input**: $T, \alpha, \beta, \pi, \pi_b, \{V_\theta|\theta\in\mathbb{R}^N\}$
**Initialization**: $\theta_0, w_0$
 1: Choose $W \sim \text{Uniform}(0,1,...,T-1)$
 2: **for** $t = 0,1,...,W-1$ **do**
 3:      Sample $O_t = (s_t, a_t, r_t, s_{t+1})$ following $\pi_b$
 4:      $\delta_t(\theta_t) = r(s_t, a_t, s_{t+1}) + \gamma V_{\theta_t}(s_{t+1}) - V_{\theta_t}(s_t)$
 5:      $\rho_t = \frac{\pi(a_t|s_t)}{\pi_b(a_t|s_t)}$
 6:      $h_t(\theta_t, \omega_t) = (\rho_t\delta_t(\theta_t) - \phi_{\theta_t}(s_t)^\top\omega_t)\nabla^2 V_{\theta_t}(s_t)\omega_t$
 7:      $\omega_{t+1} = \mathbf{\Pi}_{R_\omega}\left(\omega_t + \beta\left(-\phi_{\theta_t}(s_t)\phi_{\theta_t}(s_t)^\top\omega_t + \rho_t\delta_t(\theta_t)\phi_{\theta_t}(s_t)\right)\right)$
 8:      $\theta_{t+1} = \theta_t + \alpha\left(\rho_t\delta_t(\theta_t)\phi_{\theta_t}(s_t) - \gamma\rho_t\phi_{\theta_t}(s_{t+1})\phi_{\theta_t}(s_t)^\top\omega_t - h_t(\theta_t,\omega_t)\right)$
 9: **end for**
**Output**: $\theta_W$

---

In Algorithm 1, $\mathbf{\Pi}_{R_\omega}(v) = \arg\min_{\|w\|\le R_\omega}\|v - w\|$ denotes the projection operator, where $R_\omega = \frac{\rho_{\max}C_\phi}{\lambda_v}(r_{\max} + (1+\gamma)C_v)$ (the constants are defined in Section 3.1). As we will show in (44) in the appendix that for any $\theta\in\mathbb{R}^N$, $\omega(\theta)$ is always upper bounded by $R_\omega$, i.e., $\|\omega(\theta)\| \le R_\omega$. The projection step in the algorithm is introduced mainly for the convenience of the analysis. Motivated by the randomized stochastic gradient method in [Ghadimi and Lan, 2013], which is designed to analyze non-convex optimization problems, in this paper, we also consider a randomized version of the non-linear TDC algorithm. Specifically, let $W$ be an independent random variable with a uniform distribution over $\{0,1,...,T-1\}$. We then run the non-linear TDC algorithm for $W$ steps and output $\theta_W$.

### 3.1 Non-asymptotic Error Bounds

In this section, we present our main results of the non-asymptotic error bounds on the convergence of the off-policy non-linear TDC algorithm. Our results will be based on the following assumptions.

**Assumption 1** (Boundedness and Smoothness). *For any $s \in \mathcal{S}$ and any $\theta, \theta' \in \mathbb{R}^N$,*

$$|V_\theta(s)| \le C_v, \qquad\qquad \|\phi_\theta(s)\| \le C_\phi,$$
$$\|\nabla^2 V_\theta(s)\| \le D_v, \qquad\qquad \|\nabla^2 V_\theta(s) - \nabla^2 V_{\theta'}(s)\| \le L_V\|\theta - \theta'\|,$$

*where $C_\phi, C_v, D_v$ and $L_V$ are some positive constants.*

From Assumption 1, it follows that for any $\theta, \theta' \in \mathbb{R}^N$, $|V_\theta(s) - V_{\theta'}(s)| \le C_\phi\|\theta - \theta'\|$, and $\|\phi_\theta(s) - \phi_{\theta'}(s)\| \le D_v\|\theta - \theta'\|$. We note that these assumptions are equivalent to the assumptions adopted in the original non-linear TDC asymptotic convergence analysis in [Bhatnagar et al., 2009], and can be easily satisfied by appropriately choosing the function class $\{V_\theta : \theta \in \mathbb{R}^N\}$. For example, in neural networks, these assumptions can be satisfied if the activation function is Lipschitz and smooth [Du et al., 2019, Neyshabur, 2017, Miyato et al., 2018].

**Assumption 2** (Non-singularity). *For any $\theta \in \mathbb{R}^N$, $\lambda_L(A_\theta) \geq \lambda_v > 0$, where $\lambda_L(A)$ denotes the minimal eigenvalue of the matrix $A$ and $\lambda_v$ is a positive constant.*

**Assumption 3** (Bounded Importance Sampling Ratio). *For any $(s,a) \in \mathcal{S} \times \mathcal{A}$, $\rho(s,a) = \frac{\pi(a|s)}{\pi_b(a|s)} \leq \rho_{\max}$, for some positive constant $\rho_{\max}$.*

The following assumption is only needed for the analysis under the Markovian setting, and is widely used for analyzing the Markovian noise, e.g., [Wang and Zou, 2020, Kaledin et al., 2020, Xu and Liang, 2021, Zou et al., 2019, Srikant and Ying, 2019, Bhandari et al., 2018].

**Assumption 4** (Geometric uniform ergodicity). *There exist some constants $m > 0$ and $\kappa \in (0,1)$ such that $\sup_{s \in \mathcal{S}} d_{TV}(\mathbb{P}(s_t = \cdot|s_0 = s, \pi), \mu^\pi) \leq m\kappa^t$, for any $t > 0$, where $d_{TV}$ denotes the total-variation distance between the probability measures.*

We then present the bounds on the convergence of the TDC algorithm with general smooth function approximation in the following theorem.

**Theorem 1.** *Consider the following step-sizes: $\alpha = \mathcal{O}\left(\frac{1}{T^a}\right)$, and $\beta = \mathcal{O}\left(\frac{1}{T^b}\right)$, where $\frac{1}{2} \leq a \leq 1$ and $0 < b \leq a$. Then, (1) under the i.i.d. setting, $\|\nabla J(\theta_W)\|^2 = \mathcal{O}\left(\frac{1}{T^{1-a}} + \frac{1}{T^b} + \frac{1}{T^{1-b}}\right)$; and (2) under the Markovian setting, $\|\nabla J(\theta_W)\|^2 = \mathcal{O}\left(\frac{\log T}{T^{1-a}} + \frac{1}{T^{1-b}} + \frac{\log T}{T^b}\right)$.*

Here we only assume the order of the step-sizes in terms of $T$ for simplicity, their exact assumptions on them can be found in Section B.3 and Section C.3. Similarly, we only provide the order of the bounds here, and the explicit bounds can be found in (86) and (133) in the appendix. It can be seen that the rate under the Markovian setting is slower than the one under the i.i.d. setting by a factor of $\log T$, which is essentially the mixing time introduced by the dependence of samples.

Theorem 1 characterizes the dependence between convergence rate and the step-sizes $\alpha$ and $\beta$. We also optimize over the step-sizes in the following corollary.

**Corollary 1.** *Let $a = b = \frac{1}{2}$, i.e., $\alpha, \beta = \mathcal{O}(1/\sqrt{T})$, then (1) under the i.i.d. setting, $\|\nabla J(\theta_W)\|^2 = \mathcal{O}(1/\sqrt{T})$; and (2) under the Markovian setting, $\|\nabla J(\theta_W)\|^2 = \mathcal{O}(\log T/\sqrt{T})$.*

**Remark 1.** Our result matches with the sample complexity for the batch-based algorithm in [Xu and Liang, 2021]. But their work requires a large batch size of $\mathcal{O}(\epsilon^{-1})$ to control the bias and variance, while ours only needs one sample in each step to update $\theta$ and $\omega$ and can still obtain the same convergence rate. We note that by setting the batch size being one in [Xu and Liang, 2021], their desired sample complexity cannot be obtained, and their error bound will be a *constant*. To obtain our non-asymptotic bound and sample complexity for the non-linear TDC algorithm, we develop a novel and more refined analysis on the tracking error, which will be discussed in the next section. Moreover, our result matches with the convergence rate of solving general non-convex optimization problems using stochastic gradient descent in [Ghadimi and Lan, 2013]. Compared to their work, our analysis is more challenging due to the two time-scale structure and the gradient bias from the Markovian noise and the tracking error.

**Remark 2.** Some analyses on two time-scale stochastic approximation bound the tracking error in terms of $\frac{\alpha}{\beta}$, and require $\frac{\alpha}{\beta} \to 0$ in order to drive the tracking error to zero resulting in a convergence rate of $\mathcal{O}\left(\beta + \frac{\alpha}{\beta}\right)$ [Borkar, 2009]. In this paper, we develop a much tighter bound on the tracking error in terms of the slow time-scale parameter $\nabla J(\theta)$. Therefore, the tracking error in our analysis is driven to zero by $\nabla J(\theta) \to 0$ not $\frac{\alpha}{\beta} \to 0$. Similar results that do not need $\frac{\alpha}{\beta} \to 0$ can also be found, e.g., in [Konda et al., 2004, Kaledin et al., 2020]. We would like to point out that the techniques in [Konda et al., 2004, Kaledin et al., 2020] cannot be applied in our analysis due to the non-linear two time-scale updates in this paper.

## 4 Proof Sketch

In this section, we provide an outline of the proof of Theorem 1 under the Markovian setting, and highlight our major technical contributions. For the complete proof of Theorem 1, we refer the readers to Appendices B.2 and C.2.

Let $O_t = (s_t, a_t, r_t, s_{t+1})$ be the sample observed at time $t$. Denote the tracking error by $z_t = \omega_t - \omega(\theta_t)$, which characterizes the error between the fast time-scale update and its limit if the slow time-scale update $\theta_t$ is kept fixed and only the fast time-scale is being updated. Denote by $G_{t+1}(\theta, \omega) \triangleq \rho_t \delta_t(\theta) \phi_\theta(s_t) - \gamma \rho_t \phi_\theta(s_{t+1}) \phi_\theta(s_t)^\top \omega_t - h_t(\theta, \omega)$. Denote by $\tau_\beta$ the mixing time of the MDP, i.e., $\tau_\beta \triangleq \min \{t : m\kappa^t \leq \beta\}$.

**Step 1.** In this step, we decompose the error of gradient norm into two parts: the stochastic bias and the tracking error. We first show in Appendix A that $J(\theta)$ is $L_J$-smooth: for any $\theta_1, \theta_2 \in \mathbb{R}^N$,

$$\|\nabla J(\theta_1) - \nabla J(\theta_2)\| \leq L_J \|\theta_1 - \theta_2\|. \tag{9}$$

We note that the smoothness of $J(\theta)$ is also used in [Xu and Liang, 2021], which, however, is assumed instead of being proved as in this paper. It then follows that

$$\frac{\alpha}{2} \|\nabla J(\theta_t)\|^2 \leq J(\theta_t) - J(\theta_{t+1}) + \underbrace{\alpha \langle \nabla J(\theta_t), -G_{t+1}(\theta_t, \omega(\theta_t)) + G_{t+1}(\theta_t, \omega_t) \rangle}_{(a)}$$

$$+ \underbrace{\alpha \left\langle \nabla J(\theta_t), \frac{\nabla J(\theta_t)}{2} + G_{t+1}(\theta_t, \omega(\theta_t)) \right\rangle}_{(b)} + \frac{L_J}{2} \alpha^2 \|G_{t+1}(\theta_t, \omega_t)\|^2. \tag{10}$$

This implies that the error bound on the gradient norm is controlled by the tracking error $(a)$ which is introduced by the two time-scale update rule, and the stochastic bias $(b)$ which is due to the time-varying projection and the Markovian sampling.

**Step 2.** We first bound the tracking error. Re-write the update of $\omega_t$ in terms of $z_t$: $z_{t+1} = z_t + \beta(-A_{\theta_t}(s_t) z_t + b_t(\theta_t)) + \omega(\theta_t) - \omega(\theta_{t+1})$, where $A_{\theta_t}(s_t) = \phi_{\theta_t}(s_t) \phi_{\theta_t}(s_t)^\top$ and $b_t(\theta_t) = -A_{\theta_t}(s_t)\omega(\theta_t) + \rho_t \delta_t(\theta_t)\phi_{\theta_t}(s_t)$. From the Lipschitz continuity of $\omega(\theta)$, it follows that

$$\|z_{t+1}\| \leq (1 + \beta C_\phi^2)\|z_t\| + \beta(b_{\max} + L_\omega C_g),$$
$$\|z_{t+1} - z_t\| \leq \beta C_\phi^2 \|z_t\| + \beta(b_{\max} + L_\omega C_g), \tag{11}$$

which further implies

$$\mathbb{E}\left[\|z_{t+1}\|^2 - \|z_t\|^2\right]$$
$$\leq \underbrace{\mathbb{E}[2z_t^\top(z_{t+1} - z_t + \beta A_{\theta_t} z_t)]}_{(c)} + \mathcal{O}\left(\beta^2 \mathbb{E}[\|z_t\|^2] + \beta^2\right) + \beta \mathbb{E}\left[2z_t^\top(-A_{\theta_t})z_t\right], \tag{12}$$

where the last term in (12) can be further upper bounded by $-2\beta\lambda_v \mathbb{E}[\|z_t\|^2]$.

One challenging part in our analysis is to bound term $(c)$. Equivalently, we decompose the following term into three parts:

$$\mathbb{E}\left[z_t^\top \left(-A_{\theta_t} z_t - \frac{1}{\beta}(z_{t+1} - z_t)\right)\right]$$
$$= \underbrace{\mathbb{E}[z_t^\top(-A_{\theta_t} + A_{\theta_t}(s_t))z_t]}_{(d)} - \underbrace{\mathbb{E}[z_t^\top b_t(\theta_t)]}_{(e)} - \underbrace{\mathbb{E}\left[z_t^\top \frac{\omega(\theta_t) - \omega(\theta_{t+1})}{\beta}\right]}_{(f)}. \tag{13}$$

Consider term $(d)$ in (13). Unlike the case with linear function approximation, where the character function $\nabla V_\theta(s) = \phi(s)$ is independent with $\theta$, here the character function $\phi_\theta(s)$ depends on $\theta$. We use the geometric uniform ergodicity property of the MDP and the Lipschitz continuity of $A_\theta$ and $A_\theta(s)$ to decouple the dependence. More specifically, for any fixed $\theta$, $\mathbb{E}[A_\theta(s_t)]$ converges to $A_\theta$ as $t$ increases. Let $t = \tau_\beta$, then we have that

$$\mathbb{E}\left[z_{\tau_\beta}^\top(-A_{\theta_{\tau_\beta}} + A_{\theta_{\tau_\beta}}(s_{\tau_\beta}))z_{\tau_\beta}\right]$$
$$= \mathbb{E}\left[z_0^\top(-A_{\theta_0} + A_{\theta_0}(s_{\tau_\beta}))z_0\right] + \mathbb{E}\left[z_0^\top(-A_{\theta_{\tau_\beta}} + A_{\theta_{\tau_\beta}}(s_{\tau_\beta}) + A_{\theta_0} - A_{\theta_0}(s_{\tau_\beta}))z_0\right]$$
$$+ \mathbb{E}\left[(z_{\tau_\beta} - z_0)^\top(-A_{\theta_{\tau_\beta}} + A_{\theta_{\tau_\beta}}(s_{\tau_\beta}))(z_{\tau_\beta} - z_0)\right] + 2\mathbb{E}\left[(z_{\tau_\beta} - z_0)^\top(-A_{\theta_{\tau_\beta}} + A_{\theta_{\tau_\beta}}(s_{\tau_\beta}))z_0\right], \tag{14}$$

which can be further bounded using the mixing time $\tau_\beta$ and the Lipschitz property of $A_\theta$ and $A_\theta(s_{\tau_\beta})$. We note that from the update of $z_t$, we can bound $\|z_{\tau_\beta} - z_0\|$ and $\|z_0\|$ by $\|z_{\tau_\beta}\|$, hence the bound in (14) can be bounded in terms of $\|z_{\tau_\beta}\|$.

Similarly, note that $\mathbb{E}[b_t(\theta)]$ converges to 0 as $t \to \infty$, then we can also bound term $(e)$ in (13):

$$\mathbb{E}[z_{\tau_\beta}^\top b_{\tau_\beta}(\theta_{\tau_\beta})] = \mathbb{E}[(z_{\tau_\beta} - z_0)^\top b_{\tau_\beta}(\theta_{\tau_\beta})] + \mathbb{E}[z_0^\top b_{\tau_\beta}(\theta_0)] + \mathbb{E}[z_0^\top (b_{\tau_\beta}(\theta_{\tau_\beta}) - b_{\tau_\beta}(\theta_0))], \quad (15)$$

which can be similarly bounded in terms of $\|z_{\tau_\beta}\|$.

The challenge of bounding the third term $(f)$ in (13) lies in bounding the difference between $\omega(\theta_t)$ and $\omega(\theta_{t+1})$. One simple approach is to use the Lipschitz continuity of $\omega(\theta)$ and bound $\|\theta_t - \theta_{t+1}\|$ by a constant of order $\mathcal{O}(\alpha)$, but this will lead to a loose bound because the update $G_{t+1}(\theta_t, \omega_t)$ is actually an estimator of the gradient, which will also converge to zero. The key idea in our analysis is to bound term $(f)$ in terms of the gradient of the objective function $\nabla J(\theta)$. Specifically, we first rewrite term $\langle z_t, \omega(\theta_t) - \omega(\theta_{t+1})\rangle = -\langle z_t, \nabla\omega(\hat{\theta}_t)(\theta_{t+1} - \theta_t)\rangle = -\alpha\langle\nabla\omega(\hat{\theta}_t)G_{t+1}(\theta_t, \omega_t)\rangle$, where $\hat{\theta}_t = c\theta_t + (1-c)\theta_{t+1}$ for some $c \in [0, 1]$. It can be shown that

$$\mathbb{E}\left[z_{\tau_\beta}^\top \frac{\omega(\theta_{\tau_\beta}) - \omega(\theta_{\tau_\beta+1})}{\beta}\right] = -\frac{\alpha}{\beta}\mathbb{E}[z_{\tau_\beta}^\top \nabla\omega(\hat{\theta}_{\tau_\beta})(G_{\tau_\beta+1}(\theta_{\tau_\beta}, \omega_{\tau_\beta}) - G_{\tau_\beta+1}(\theta_{\tau_\beta}, \omega(\theta_{\tau_\beta})))]$$

$$- \frac{\alpha}{\beta}\mathbb{E}\left[z_{\tau_\beta}^\top \nabla\omega(\hat{\theta}_{\tau_\beta})\left(G_{\tau_\beta+1}(\theta_{\tau_\beta}, \omega(\theta_{\tau_\beta})) + \frac{\nabla J(\theta_{\tau_\beta})}{2}\right)\right] + \frac{\alpha}{\beta}\mathbb{E}\left[z_{\tau_\beta}^\top \nabla\omega(\hat{\theta}_{\tau_\beta})\left(\frac{\nabla J(\theta_{\tau_\beta})}{2}\right)\right]. \quad (16)$$

The first term in (16) can be bounded in terms of $\|z_{\tau_\beta}\|^2$ using the Lipschitz property of $G_{\tau_\beta+1}(\theta, \omega)$ in $\omega$. The second term can be bounded using the uniform ergodicity of the MDP and the Lipschitz property of $z_0^\top \nabla\omega(\theta)\left(G_{\tau_\beta+1}(\theta, \omega(\theta)) + \frac{\nabla J(\theta)}{2}\right)$ in $\theta$. The third term can be bounded in terms of $\|z_{\tau_\beta}\|^2$ and $\|\nabla J(\theta_{\tau_\beta})\|^2$. Combining all bounds together, we have the bound on term $(f)$ in (13):

$$\left|\mathbb{E}\left[z_{\tau_\beta}^\top \frac{\omega(\theta_{\tau_\beta}) - \omega(\theta_{\tau_\beta+1})}{\beta}\right]\right|$$

$$\leq \mathcal{O}\left(\frac{\alpha}{\beta}\right)\mathbb{E}\left[\|z_{\tau_\beta}\|^2\right] + \mathcal{O}(\alpha\tau_\beta)\mathbb{E}\left[\|z_{\tau_\beta}\|\right] + \mathcal{O}(\alpha\tau_\beta) + \mathcal{O}\left(\frac{\alpha}{8\beta}\right)\mathbb{E}\left[\|\nabla J(\theta_{\tau_\beta})\|^2\right]. \quad (17)$$

We combine all the bounds on terms $(d), (e)$ and $(f)$ and hence get the error bound on (13):

$$\mathbb{E}\left[z_t^\top\left(-A_{\theta_t}z_t - \frac{1}{\beta}(z_{t+1} - z_t)\right)\right] \leq \mathcal{O}\left(\frac{\alpha}{\beta}\right)\mathbb{E}[\|z_t\|^2] + \mathcal{O}(\beta\tau_\beta) + \mathcal{O}\left(\frac{\alpha}{\beta}\right)\mathbb{E}[\|\nabla J(\theta_t)\|^2]. \quad (18)$$

Plugging the above bound in (12), we have the following recursive bound on the tracking error:

$$\mathbb{E}\left[\|z_{t+1}\|^2\right] \leq \mathcal{O}(1 - \beta)\mathbb{E}\left[\|z_t\|^2\right] + \mathcal{O}(\alpha)\mathbb{E}\left[\|\nabla J(\theta_t)\|^2\right] + \mathcal{O}(\beta^2\tau_\beta). \quad (19)$$

Then by recursively applying the inequality in (19) and summing up w.r.t. $t$ from 0 to $T-1$, we obtain the bound on the tracking error $\sum_{t=0}^{T-1}\mathbb{E}[\|z_t\|^2]/T$:

$$\frac{\sum_{t=0}^{T-1}\mathbb{E}[\|z_t\|^2]}{T} \leq \mathcal{O}\left(\frac{1}{T\beta} + \frac{\alpha}{\beta}\frac{\sum_{t=0}^{T-1}\mathbb{E}[\|\nabla J(\theta_t)\|^2]}{T} + \beta\tau_\beta\right).$$

**Step 3.** In this step we bound the stochastic bias term $\mathbb{E}\left[\left\langle\nabla J(\theta_t), \frac{\nabla J(\theta_t)}{2} + G_{t+1}(\theta_t, \omega(\theta_t))\right\rangle\right]$. Similarly, we add and subtract $\nabla J(\theta_0)$ and $G_{\tau_\beta+1}(\theta_0, \omega(\theta_0))$, and obtain that

$$\left\langle\nabla J(\theta_{\tau_\beta}), \frac{\nabla J(\theta_{\tau_\beta})}{2} + G_{\tau_\beta+1}(\theta_{\tau_\beta}, \omega(\theta_{\tau_\beta}))\right\rangle$$

$$= \left\langle\nabla J(\theta_0), \frac{\nabla J(\theta_0)}{2} + G_{\tau_\beta+1}(\theta_0, \omega(\theta_0))\right\rangle + \left(\left\langle\nabla J(\theta_{\tau_\beta}), \frac{\nabla J(\theta_{\tau_\beta})}{2} + G_{\tau_\beta+1}(\theta_{\tau_\beta}, \omega(\theta_{\tau_\beta}))\right\rangle\right.$$

$$- \left\langle \nabla J(\theta_0), \frac{\nabla J(\theta_0)}{2} + G_{\tau_\beta+1}(\theta_0, \omega(\theta_0)) \right\rangle \Bigg), \tag{20}$$

which again can be bounded using the geometry uniform ergodicity of the MDP and the Lipschitz continuity of $\left\langle \nabla J(\theta), \frac{\nabla J(\theta)}{2} + G_{\tau_\beta+1}(\theta, \omega(\theta)) \right\rangle$.

**Step 4.** Plugging in the bounds on the tracking error and the stochastic bias and rearranging the terms, then it follows that $\frac{\sum_{t=0}^{T-1} \mathbb{E}[\|\nabla J(\theta_t)\|^2]}{T} \leq U \sqrt{\frac{\sum_{t=0}^{T-1} \mathbb{E}[\|\nabla J(\theta_t)\|^2]}{T}} + V$, where $U$ and $V$ are some constants depending on the step sizes, and the explicit definitions can be found in (132). By solving the inequality of $\frac{\sum_{t=0}^{T-1} \mathbb{E}[\|\nabla J(\theta_t)\|^2]}{T}$, we obtain that

$$\frac{\sum_{t=0}^{T-1} \mathbb{E}[\|\nabla J(\theta_t)\|^2]}{T} \leq \mathcal{O}\left( \beta\tau_\beta + \frac{1}{T\beta} + \alpha\tau_\beta + \frac{1}{T\alpha} \right).$$

## 5 Conclusion

In this paper, we extend the on-policy non-linear TDC algorithm to the off-policy setting, and characterize its non-asymptotic error bounds under both the i.i.d. and the Markovian settings. We show that the non-linear TDC algorithm converges as fast as $\mathcal{O}(1/\sqrt{T})$ (up to a factor of $\log T$). The techniques and tools developed in this paper can be used to analyze a wide range of value-based RL algorithms with general smooth function approximation.

**Limitations:** It is not clear yet whether the stationary points that the TDC converges to are second-order stationary or potentially saddle points.

**Negative social impacts:** This work is a theoretical investigation of some fundamental RL algorithms, and therefore, the authors do not foresee any negative societal impact.

## 6 Acknowledgment

The work of Yue Wang and Shaofeng Zou was supported in part by the National Science Foundation under Grants CCF-2106560 and CCF-2007783. Yi Zhou's work was supported in part by U.S. National Science Foundation under the Grant CCF-2106216.

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
