# Part

# Appendix

## Table of Contents

We first introduce some notations. In the following proofs, $\|a\|$ denotes the $\ell_2$ norm if $a$ is a vector; and $\|A\|$ denotes the operator norm if $A$ is a matrix.

In Appendix A, we prove the Lipschitz continuity of some important functions, including $\omega(\theta)$, $\nabla\omega(\theta)$ and the gradient $\nabla J(\theta)$ of objective function. In Appendix B, we present the non-asymptotic analysis for the i.i.d. setting. In Appendix C, we present the non-asymptotic analysis for the Markovian setting. In appendix D, we present some numerical experiments.

## A  Useful Lemmas

### A.1  Lipschitz Continuity of $\omega(\theta)$

In this section, we show that $\omega(\theta)$ is Lipschitz in $\theta$.

**Lemma 1.** *For any $\theta, \theta' \in \mathbb{R}^N$, we have that*

$$\|\omega(\theta) - \omega(\theta')\| \le L_\omega \|\theta - \theta'\|, \tag{21}$$

*where $L_\omega = \frac{1}{\lambda_v}\left((1+\gamma)C_\phi^2 + (r_{\max} + (1+\gamma)C_v)D_v\right) + \frac{2C_\phi^2 D_v}{\lambda_v^2}(r_{\max} + (1+\gamma)C_v)$.*

*Proof.* Recall that

$$\begin{aligned}
\omega(\theta) &= \mathbb{E}_{\mu^{\pi_b}}[\phi_\theta(S)\phi_\theta(S)^\top]^{-1}\mathbb{E}_{\mu^{\pi_b}}[\rho(S,A)\delta_{S,A,S'}(\theta)\phi_\theta(S)] \\
&= A_\theta^{-1}\mathbb{E}_{\mu^{\pi_b}}[\rho(S,A)\delta_{S,A,S'}(\theta)\phi_\theta(S)], \tag{22}
\end{aligned}$$

hence we can show the conclusion by showing that $A_\theta^{-1}$ and $\mathbb{E}_{\mu^{\pi_b}}[\rho(S,A)\delta_{S,A,S'}(\theta)\phi_\theta(S)]$ are both Lipschitz and bounded.

From Assumption 2, we know that

$$\|A_\theta^{-1}\| \le \frac{1}{\lambda_v}. \tag{23}$$

We also show that

$$
\begin{aligned}
&\|A_\theta^{-1} - A_{\theta'}^{-1}\| \\
&= \|A_\theta^{-1} A_{\theta'} A_{\theta'}^{-1} - A_\theta^{-1} A_\theta A_{\theta'}^{-1}\| \\
&= \|A_\theta^{-1}(A_{\theta'} - A_\theta) A_{\theta'}^{-1}\| \\
&\leq \frac{2C_\phi D_v}{\lambda_v^2} \|\theta - \theta'\|,
\end{aligned} \tag{24}
$$

which is from the fact that $\|A_\theta - A_{\theta'}\| = \|\mathbb{E}_{\mu^{\pi_b}}[\phi_\theta(S)\phi_\theta(S)^\top] - \mathbb{E}_{\mu^{\pi_b}}[\phi_{\theta'}(S)\phi_{\theta'}(S)^\top]\| \leq 2C_\phi D_v \|\theta - \theta'\|$.

By Assumption 1 and the boundedness of the reward function, it can be shown that for any $\theta \in \mathbb{R}^N$ and any $(s, a, s') \in \mathcal{S} \times \mathcal{A} \times \mathcal{S}$,

$$
|\delta_{s,a,s'}(\theta)| = |r(s, a, s') + \gamma V_\theta(s') - V_\theta(s)| \leq r_{\max} + (1 + \gamma)C_v. \tag{25}
$$

We then show that $\delta_{s,a,s'}(\theta)$ is Lipschitz, i.e., for any $\theta, \theta' \in \mathbb{R}^N$ and any $(s, a, s') \in \mathcal{S} \times \mathcal{A} \times \mathcal{S}$,

$$
\begin{aligned}
&|\delta_{s,a,s'}(\theta) - \delta_{s,a,s'}(\theta')| \\
&= |\gamma V_\theta(s') - V_\theta(s) - \gamma V_{\theta'}(s') - V_{\theta'}(s)| \\
&\leq (\gamma + 1)C_\phi \|\theta - \theta'\|.
\end{aligned} \tag{26}
$$

Hence, the function $\|\mathbb{E}_{\mu^{\pi_b}}[\rho(S, A)\delta_{S,A,S'}(\theta)\phi_\theta(S)]\|$ is Lipschitz:

$$
\begin{aligned}
&\|\mathbb{E}_{\mu^{\pi_b}}[\rho(S, A)\delta_{S,A,S'}(\theta)\phi_\theta(S)] - \mathbb{E}_{\mu^{\pi_b}}[\rho(S, A)\delta_{S,A,S'}(\theta')\phi_{\theta'}(S)]\| \\
&= \|\mathbb{E}_{\mu^{\pi_b}}[\rho(S, A)\delta_{S,A,S'}(\theta)\phi_\theta(S)] - \mathbb{E}_{\mu^{\pi_b}}[\rho(S, A)\delta_{S,A,S'}(\theta')\phi_\theta(S)] \\
&\quad + \mathbb{E}_{\mu^{\pi_b}}[\rho(S, A)\delta_{S,A,S'}(\theta')\phi_\theta(S)] - \mathbb{E}_{\mu^{\pi_b}}[\rho(S, A)\delta_{S,A,S'}(\theta')\phi_{\theta'}(S)]\| \\
&\leq \|\mathbb{E}_{\mu^{\pi_b}}[\rho(S, A)\delta_{S,A,S'}(\theta)\phi_\theta(S)] - \mathbb{E}_{\mu^{\pi_b}}[\rho(S, A)\delta_{S,A,S'}(\theta')\phi_\theta(S)]\| \\
&\quad + \|\mathbb{E}_{\mu^{\pi_b}}[\rho(S, A)\delta_{S,A,S'}(\theta')\phi_\theta(S)] - \mathbb{E}_{\mu^{\pi_b}}[\rho(S, A)\delta_{S,A,S'}(\theta')\phi_{\theta'}(S)]\| \\
&\leq \mathbb{E}_{\mu^{\pi_b}}[\rho(S, A)|\delta_{S,A,S'}(\theta) - \delta_{S,A,S'}(\theta')|\|\phi_\theta(S)\|] \\
&\quad + \mathbb{E}_{\mu^{\pi_b}}[\rho(S, A)|\delta_{S,A,S'}(\theta')|\|\phi_\theta(S) - \phi_{\theta'}(S)\|] \\
&\overset{(a)}{\leq} (1 + \gamma)C_\phi^2 \|\theta - \theta'\| + (r_{\max} + (1 + \gamma)C_v)D_v \|\theta - \theta'\| \\
&= \left((1 + \gamma)C_\phi^2 + (r_{\max} + (1 + \gamma)C_v)D_v\right) \|\theta - \theta'\|,
\end{aligned} \tag{27}
$$

where $(a)$ is from (26) and the fact that $\mathbb{E}_{\mu^{\pi_b}}[\rho(S, A)] = 1$. Also $\|\mathbb{E}_{\mu^{\pi_b}}[\rho(S, A)\delta_{S,A,S'}(\theta)\phi_\theta(S)]\|$ can be upper bounded as follows:

$$
\|\mathbb{E}_{\mu^{\pi_b}}[\rho(S, A)\delta_{S,A,S'}(\theta)\phi_\theta(S)]\| \leq C_\phi(r_{\max} + (1 + \gamma)C_v). \tag{28}
$$

Combining (23), (24), (28) and (27), we show that $\omega(\cdot)$ is Lipschitz in $\theta$:

$$
\begin{aligned}
&\|\omega(\theta) - \omega(\theta')\| \\
&\leq \left( \frac{1}{\lambda_v} \left((1 + \gamma)C_\phi^2 + (r_{\max} + (1 + \gamma)C_v)D_v\right) + \frac{2C_\phi^2 D_v}{\lambda_v^2}(r_{\max} + (1 + \gamma)C_v) \right) \|\theta - \theta'\| \\
&\triangleq L_\omega \|\theta - \theta'\|,
\end{aligned} \tag{29}
$$

where $L_\omega = \frac{1}{\lambda_v} \left((1 + \gamma)C_\phi^2 + (r_{\max} + (1 + \gamma)C_v)D_v\right) + \frac{2C_\phi^2 D_v}{\lambda_v^2}(r_{\max} + (1 + \gamma)C_v)$. $\quad\square$

### A.2 Lipschitz Continuity of $\nabla\omega(\theta)$

In this section, we show that $\nabla\omega(\theta)$ is Lipschitz.

**Lemma 2.** *For any $\theta, \theta' \in \mathbb{R}^N$, it follows that*

$$
\|\nabla\omega(\theta) - \nabla\omega(\theta')\| \leq D_\omega \|\theta - \theta'\|, \tag{30}
$$

*where*

$$D_\omega = \left( \frac{(C_\phi L_v + 2D_v^2 + D_v C_\phi)}{\lambda_v^2} + \frac{8C_\phi^2 D_v^2}{\lambda_v^3} \right) C_\phi(r_{\max} + C_v + \gamma C_v)$$

$$+ \frac{4C_\phi D_v}{\lambda_v^2} \left( C_\phi^2(1+\gamma) + D_v(r_{\max} + (1+\gamma)C_v) \right)$$

$$+ \frac{3C_\phi D_v(1+\gamma) + L_v(r_{\max} + (1+\gamma)C_v)}{\lambda_v}. \tag{31}$$

*Proof.* Recall the definition of $\omega(\theta) = A_\theta^{-1} \mathbb{E}_{\mu^{\pi_b}}[\rho(S,A)\delta_{S,A,S'}(\theta)\phi_\theta(S)]$, hence we have

$$\nabla \omega(\theta) = -A_\theta^{-1}(\nabla A_\theta)A_\theta^{-1}\mathbb{E}_{\mu^{\pi_b}}[\rho(S,A)\delta_{S,A,S'}(\theta)\phi_\theta(S)]$$
$$+ A_\theta^{-1}\mathbb{E}_{\mu^{\pi_b}}[\nabla\rho(S,A)\delta_{S,A,S'}(\theta)\phi_\theta(S)], \tag{32}$$

where the tensor $\nabla A_\theta$ can be equivalently viewed as an operator: $\mathbb{R}^N \to \mathbb{R}^{N \times N}$, i.e., $\nabla A_\theta(w) = \nabla(A_\theta w)$ for any $w \in \mathbb{R}^N$.

We show that the operator norm of $\nabla A_\theta$ is bounded as follows:

$$\|\nabla A_\theta\| = \sup_{\|w\|=1} \|\nabla A_\theta(w)\|$$
$$= \sup_{\|w\|=1} \|\nabla(A_\theta w)\|$$
$$= \sup_{\|w\|=1} \|\nabla \mathbb{E}_{\mu^{\pi_b}}[\phi_\theta(S)\phi_\theta(S)^\top w]\|$$
$$= \sup_{\|w\|=1} \|\mathbb{E}_{\mu^{\pi_b}}[(\phi_\theta(S)^\top w)\nabla\phi_\theta(S)] + \mathbb{E}_{\mu^{\pi_b}}[\phi_\theta(S)(\nabla\phi_\theta(S)^\top w)^\top]\|$$
$$\leq \sup_{\|w\|=1} 2C_\phi D_v \|w\|$$
$$= 2C_\phi D_v. \tag{33}$$

The Lipschitz continuous of $\nabla A_\theta$ can be shown as follows:

$$\|\nabla A_\theta - \nabla A_{\theta'}\|$$
$$= \sup_{\|w\|=1} \|\nabla(A_\theta w) - \nabla(A_{\theta'} w)\|$$
$$= \sup_{\|w\|=1} \|\mathbb{E}_{\mu^{\pi_b}}[\nabla\phi_\theta(S)(\phi_\theta(S)^\top w) + (\nabla\phi_\theta(S)^\top w)\phi_\theta(S)^\top - \nabla\phi_{\theta'}(S)(\phi_{\theta'}(S)^\top w)$$
$$- (\nabla\phi_{\theta'}(S)^\top w)\phi_{\theta'}(S)^\top]\|$$
$$\leq \sup_{\|w\|=1} (C_\phi L_v + 2D_v^2 + D_v C_\phi)\|\theta - \theta'\|\|w\|$$
$$= (C_\phi L_v + 2D_v^2 + D_v C_\phi)\|\theta - \theta'\|. \tag{34}$$

Then we conclude that the operator norm of $-A_\theta^{-1}(\nabla A_\theta)$ is upper bounded by $\frac{2C_\phi D_v}{\lambda_v}$, and is Lipschitz with constant $\frac{(C_\phi L_v + 2D_v^2 + D_v C_\phi)}{\lambda_v} + \frac{4C_\phi^2 D_v^2}{\lambda_v^2}$. It can be further seen that $-A_\theta^{-1}(\nabla A_\theta)A_\theta^{-1}$ is upper bounded by $\frac{2C_\phi D_v}{\lambda_v^2}$, and Lipschitz with constant $\frac{(C_\phi L_v + 2D_v^2 + D_v C_\phi)}{\lambda_v^2} + \frac{8C_\phi^2 D_v^2}{\lambda_v^3}$.

Recall that we have shown in (28) that

$$\|\mathbb{E}_{\mu^{\pi_b}}[\rho(S,A)\delta_{S,A,S'}(\theta)\phi_\theta(S)] - \mathbb{E}_{\mu^{\pi_b}}[\rho(S,A)\delta_{S,A,S'}(\theta')\phi_{\theta'}(S)]\|$$
$$\leq \left((1+\gamma)C_\phi^2 + (r_{\max} + (1+\gamma)C_v)D_v\right)\|\theta - \theta'\|, \tag{35}$$

and it is upper bounded by $C_\phi(r_{\max} + (1+\gamma)C_V)$. Hence we have that $-A_\theta^{-1}(\nabla A_\theta)A_\theta^{-1}\mathbb{E}_{\mu^{\pi_b}}[\rho(S,A)\delta_{S,A,S'}(\theta)\phi_\theta(S)]$ can be upper bounded by $(r_{\max} + (1+\gamma)C_V)\frac{2C_\phi^2 D_v}{\lambda_v^2}$, and it is Lipschitz with constant $\left(\frac{(C_\phi L_v + 2D_v^2 + D_v C_\phi)}{\lambda_v^2} + \frac{8C_\phi^2 D_v^2}{\lambda_v^3}\right)C_\phi(r_{\max} + C_v + \gamma C_v) + \frac{2C_\phi D_v}{\lambda_v^2}\left((1+\gamma)C_\phi^2 + (r_{\max} + (1+\gamma)C_v)D_v\right) \triangleq L_A$.

For the second term of (32), we also show it is Lipschitz as follows. First note that $\nabla \delta_{s,a,s'}(\theta)\phi_\theta(s) = \nabla \delta_{s,a,s'}(\theta)\phi_\theta(s)^\top + \delta_{s,a,s'}(\theta)\nabla\phi_\theta(s)$, hence we know $\mathbb{E}_{\mu^{\pi_b}}[\nabla\rho(S,A)\delta_{S,A,S'}(\theta)\phi_\theta(S)]$ can be upper bounded by $C_\phi^2(1+\gamma) + D_v(r_{\max} + (1+\gamma)C_v)$, and is Lipschitz with constant $3C_\phi D_v(1+\gamma) + L_v(r_{\max} + (1+\gamma)C_v)$. Finally we conclude that the second term in (32) $A_\theta^{-1}\mathbb{E}_{\mu^{\pi_b}}[\nabla\rho(S,A)\delta_{S,A,S'}(\theta)\phi_\theta(S)]$ is Lipschitz with constant $\frac{2C_\phi D_v}{\lambda_v^2}\left(C_\phi^2(1+\gamma) + D_v(r_{\max} + (1+\gamma)C_v)\right) + \frac{3C_\phi D_v(1+\gamma) + L_v(r_{\max}+(1+\gamma)C_v)}{\lambda_v} \triangleq L_A'$.

Hence $\nabla\omega(\theta)$ is Lipschitz with constant $L_A + L_A' \triangleq D_\omega$, where

$$
\begin{aligned}
D_\omega = {} & \left(\frac{(C_\phi L_v + 2D_v^2 + D_v C_\phi)}{\lambda_v^2} + \frac{8C_\phi^2 D_v^2}{\lambda_v^3}\right) C_\phi(r_{\max} + C_v + \gamma C_v) \\
& + \frac{4C_\phi D_v}{\lambda_v^2}\left(C_\phi^2(1+\gamma) + D_v(r_{\max} + (1+\gamma)C_v)\right) \\
& + \frac{3C_\phi D_v(1+\gamma) + L_v(r_{\max} + (1+\gamma)C_v)}{\lambda_v}.
\end{aligned}
\tag{36}
$$

$\square$

### A.3   Smoothness of $J(\theta)$

In the following lemma, we show that the objective function $J(\theta)$ is $L_J$-smooth. We note that the smoothness of $J(\theta)$ is assumed in [Xu and Liang, 2021] instead of being proved as in this paper.

**Lemma 3.** $J(\theta)$ is $L_J$-smooth, i.e., for any $\theta, \theta' \in \mathbb{R}^N$,

$$
\|\nabla J(\theta) - \nabla J(\theta')\| \le L_J\|\theta - \theta'\|,
\tag{37}
$$

*where*

$$
\begin{aligned}
L_J = {} & 2\left((1+\gamma)C_\phi^2 + (r_{\max} + (1+\gamma)C_v)D_v\right) + 2\gamma\left(C_\phi^2 L_\omega + 2D_v\frac{C_\phi^2}{\lambda_v}(r_{\max} + (1+\gamma)C_v)\right) \\
& + 2\bigg((D_v R_\omega + C_\phi L_\omega + (1+\gamma)C_\phi)D_v R_\omega \\
& + (R_\omega L_V + D_v L_\omega)((r_{\max} + (1+\gamma)C_v) + C_\phi R_\omega)\bigg).
\end{aligned}
\tag{38}
$$

*Proof.* Before we prove the main statement, we first drive some boundedness and Lipschitz properties. Recall that

$$
\begin{aligned}
-\frac{\nabla J(\theta)}{2} = {} & \mathbb{E}_{\mu^{\pi_b}}\bigg[\left(\rho(S,A)\delta_{S,A,S'}(\theta)\phi_\theta(S) - \gamma\rho(S,A)\phi_\theta(S')\phi_\theta(S)^\top\omega(\theta)\right. \\
& \left. - h_{S,A,S'}(\theta, \omega(\theta))\right)\bigg],
\end{aligned}
\tag{39}
$$

$$
\omega(\theta) = \mathbb{E}_{\mu^{\pi_b}}[\phi_\theta(S)\phi_\theta(S)^\top]^{-1}\mathbb{E}_{\mu^{\pi_b}}[\rho(S,A)\delta_{S,A,S'}(\theta)\phi_\theta(S)],
\tag{40}
$$

$$
h_{s,a,s'}(\theta, \omega(\theta)) = (\rho(s,a)\delta_{s,a,s'}(\theta) - \phi_\theta(s)^\top\omega(\theta))\nabla^2 V_\theta(s)\omega(\theta).
\tag{41}
$$

We have shown in Lemma 1 that for any $\theta \in \mathbb{R}^N$ and any $(s,a,s') \in \mathcal{S} \times \mathcal{A} \times \mathcal{S}$,

$$
|\delta_{s,a,s'}(\theta)| = |r(s,a,s') + \gamma V_\theta(s') - V_\theta(s)| \le r_{\max} + (1+\gamma)C_v;
\tag{42}
$$

and that

$$
\begin{aligned}
& \|\mathbb{E}_{\mu^{\pi_b}}[\rho(S,A)\delta_{S,A,S'}(\theta)\phi_\theta(S)] - \mathbb{E}_{\mu^{\pi_b}}[\rho(S,A)\delta_{S,A,S'}(\theta')\phi_{\theta'}(S)]\| \\
& \le \left((1+\gamma)C_\phi^2 + (r_{\max} + (1+\gamma)C_v)D_v\right)\|\theta - \theta'\|.
\end{aligned}
\tag{43}
$$

Also it is easy to see from the definition that

$$
\|\omega(\theta)\| \le \frac{C_\phi}{\lambda_v}(r_{\max} + (1+\gamma)C_v) \triangleq R_\omega.
\tag{44}
$$

Hence the Lipschitz continuity of $\mathbb{E}_{\mu^{\pi_b}}[\rho(S,A)\phi_\theta(S')\phi_\theta(S)^\top]\omega(\theta)$ can be shown as follows

$$\left\|\mathbb{E}_{\mu^{\pi_b}}[\rho(S,A)\phi_\theta(S')\phi_\theta(S)^\top]\omega(\theta) - \mathbb{E}_{\mu^{\pi_b}}[\rho(S,A)\phi_{\theta'}(S')\phi_{\theta'}(S)^\top]\omega(\theta')\right\|$$

$$\leq \left\|\mathbb{E}_{\mu^{\pi_b}}[\rho(S,A)\phi_\theta(S')\phi_\theta(S)^\top]\omega(\theta) - \mathbb{E}_{\mu^{\pi_b}}[\rho(S,A)\phi_\theta(S')\phi_\theta(S)^\top]\omega(\theta')\right\|$$

$$+ \left\|\mathbb{E}_{\mu^{\pi_b}}[\rho(S,A)\phi_\theta(S')\phi_\theta(S)^\top]\omega(\theta') - \mathbb{E}_{\mu^{\pi_b}}[\rho(S,A)\phi_{\theta'}(S')\phi_{\theta'}(S)^\top]\omega(\theta')\right\|$$

$$\overset{(a)}{\leq} C_\phi^2 L_\omega\|\theta - \theta'\| + 2C_\phi D_v R_\omega\|\theta - \theta'\|$$

$$= \left(C_\phi^2 L_\omega + 2D_v\frac{C_\phi^2}{\lambda_v}(r_{\max} + (1+\gamma)C_v)\right)\|\theta - \theta'\|, \tag{45}$$

where $(a)$ is due to the fact that $\omega(\theta)$ is Lipschitz in (21) and the fact that

$$\left\|\mathbb{E}_{\mu^{\pi_b}}[\rho(S,A)\phi_\theta(S')\phi_\theta(S)^\top] - \mathbb{E}_{\mu^{\pi_b}}[\rho(S,A)\phi_{\theta'}(S')\phi_{\theta'}(S)^\top]\right\| \leq 2C_\phi D_v\|\theta - \theta'\|. \tag{46}$$

We then show that the function $h_{s,a,s'}(\theta, \omega(\theta))$ is Lipschitz in $\theta$ as follows. We first note that for any $s \in \mathcal{S}$ and $\theta, \theta' \in \mathbb{R}^N$,

$$\|\phi_\theta(s)^\top\omega(\theta) - \phi_{\theta'}(s)^\top\omega(\theta')\|$$

$$\leq \|\phi_\theta(s)^\top\omega(\theta) - \phi_{\theta'}(s)^\top\omega(\theta)\| + \|\phi_{\theta'}(s)^\top\omega(\theta) - \phi_{\theta'}(s)^\top\omega(\theta')\|$$

$$\leq (D_v R_\omega + C_\phi L_\omega)\|\theta - \theta'\|. \tag{47}$$

This implies that for any $(s, a, s') \in \mathcal{S} \times \mathcal{A} \times \mathcal{S}$ and $\theta, \theta' \in \mathbb{R}^N$,

$$\|\rho(s,a)\delta_{s,a,s'}(\theta) - \phi_\theta(s)^\top\omega(\theta) - \rho(s,a)\delta_{s,a,s'}(\theta') + \phi_{\theta'}(s)^\top\omega(\theta')\|$$

$$\leq (D_v R_\omega + C_\phi L_\omega + (1+\gamma)C_\phi\rho(s,a))\|\theta - \theta'\|. \tag{48}$$

We also show the following function is Lipschitz:

$$\|\nabla^2 V_\theta(s)\omega(\theta) - \nabla^2 V_{\theta'}(s)\omega(\theta')\|$$

$$\leq \|\nabla^2 V_\theta(s)\omega(\theta) - \nabla^2 V_{\theta'}(s)\omega(\theta)\| + \|\nabla^2 V_{\theta'}(s)\omega(\theta) - \nabla^2 V_{\theta'}(s)\omega(\theta')\|$$

$$\leq R_\omega L_V\|\theta - \theta'\| + D_v L_\omega\|\theta - \theta'\|$$

$$= (R_\omega L_V + D_v L_\omega)\|\theta - \theta'\|. \tag{49}$$

Combining (48) and (49), it can be shown that $h_{s,a,s'}(\theta, \omega(\theta))$ is Lipschitz in $\theta$ as follows

$$\|h_{s,a,s'}(\theta, \omega(\theta)) - h_{s,a,s'}(\theta', \omega(\theta'))\|$$

$$= \|\left(\rho(s,a)\delta_{s,a,s'}(\theta) - \phi_\theta(s)^\top\omega(\theta)\right)\nabla^2 V_\theta(s)\omega(\theta)$$

$$- \left(\rho(s,a)\delta_{s,a,s'}(\theta') - \phi_{\theta'}(s)^\top\omega(\theta')\right)\nabla^2 V_{\theta'}(s)\omega(\theta')\|$$

$$\leq ((D_v R_\omega + C_\phi L_\omega + (1+\gamma)C_\phi\rho(s,a))D_v R_\omega)\|\theta - \theta'\|$$

$$+ (R_\omega L_V + D_v L_\omega)(\rho(s,a)(r_{\max} + (1+\gamma)C_v) + C_\phi R_\omega)\|\theta - \theta'\|. \tag{50}$$

From the results in (43), (45) and (50), it follows that

$$\|\nabla J(\theta) - \nabla J(\theta')\|$$

$$\leq 2\left\|\mathbb{E}_{\mu^{\pi_b}}[\rho(S,A)\delta_{S,A,S'}(\theta)\phi_\theta(S) - \rho(S,A)\delta_{S,A,S'}(\theta')\phi_{\theta'}(S)]\right\|$$

$$+ 2\gamma\left\|\mathbb{E}_{\mu^{\pi_b}}[\rho(S,A)\phi_\theta(S')\phi_\theta(S)^\top\omega(\theta) - \rho(S,A)\phi_{\theta'}(S')\phi_{\theta'}(S)^\top\omega(\theta')]\right\|$$

$$+ 2\left\|\mathbb{E}_{\mu^{\pi_b}}[h_{S,A,S'}(\theta, \omega(\theta)) - h_{S,A,S'}(\theta', \omega(\theta'))]\right\|$$

$$\leq 2\left((1+\gamma)C_\phi^2 + (r_{\max} + (1+\gamma)C_v)D_v\right)\|\theta - \theta'\|$$

$$+ 2\gamma\left(C_\phi^2 L_\omega + 2D_v\frac{C_\phi^2}{\lambda_v}(r_{\max} + (1+\gamma)C_v)\right)\|\theta - \theta'\|$$

$$+ 2\mathbb{E}_{\mu^{\pi_b}}[((D_v R_\omega + C_\phi L_\omega + (1+\gamma)C_\phi\rho(S,A))D_v R_\omega)]\|\theta - \theta'\|$$

$$+ 2\mathbb{E}_{\mu^{\pi_b}}[(R_\omega L_V + D_v L_\omega)(\rho(S,A)(r_{\max} + (1+\gamma)C_v) + C_\phi R_\omega)]\|\theta - \theta'\|$$

$$
\overset{(a)}{\leq} 2\left((1+\gamma)C_\phi^2 + (r_{\max} + (1+\gamma)C_v)D_v\right)\|\theta - \theta'\|
$$

$$
+ 2\gamma\left(C_\phi^2 L_\omega + 2D_v \frac{C_\phi^2}{\lambda_v}(r_{\max} + (1+\gamma)C_v)\right)\|\theta - \theta'\|
$$

$$
+ 2\big((D_v R_\omega + C_\phi L_\omega + (1+\gamma)C_\phi)D_v R_\omega
$$

$$
+ (R_\omega L_V + D_v L_\omega)((r_{\max} + (1+\gamma)C_v) + C_\phi R_\omega)\big)\|\theta - \theta'\|
$$

$$
\triangleq L_J\|\theta - \theta'\|, \tag{51}
$$

where $(a)$ is due to the fact that $\mathbb{E}_{\mu^{\pi_b}}[\rho(S,A)] = 1$, and

$$
L_J = 2\left((1+\gamma)C_\phi^2 + (r_{\max} + (1+\gamma)C_v)D_v\right) + 2\gamma\left(C_\phi^2 L_\omega + 2D_v\frac{C_\phi^2}{\lambda_v}(r_{\max} + (1+\gamma)C_v)\right)
$$

$$
+ 2\big((D_v R_\omega + C_\phi L_\omega + (1+\gamma)C_\phi)D_v R_\omega
$$

$$
+ (R_\omega L_V + D_v L_\omega)((r_{\max} + (1+\gamma)C_v) + C_\phi R_\omega)\big). \tag{52}
$$

This completes the proof. $\qquad\square$

## B  Non-asymptotic Analysis under the i.i.d. Setting

First we introduce the off-policy TDC learning with non-linear function approximation algorithm under the i.i.d. setting in Algorithm 2. We then bound the tracking error in Appendix B.1, and prove the Theorem 1 under the i.i.d. setting in Appendix B.2.

---
**Algorithm 2** Non-Linear Off-Policy TDC under the i.i.d. Setting
---
**Input**: $T, \alpha, \beta, \pi, \pi_b, \{V_\theta | \theta \in \mathbb{R}^N\}$
**Initialization**: $\theta_0, \omega_0$
  1: Choose $W \sim \text{Uniform}(0, 1, ..., T-1)$
  2: **for** $t = 0, 1, ..., W-1$ **do**
  3:     Sample $O_t = (s_t, a_t, r_t, s_t')$ according to $\mu^{\pi_b}$
  4:     $\rho_t = \frac{\pi(a_t|s_t)}{\pi_b(a_t|s_t)}$
  5:     $\delta_t(\theta_t) = r(s_t, a_t, s_t') + \gamma V_{\theta_t}(s_t') - V_{\theta_t}(s_t)$
  6:     $h_t(\theta_t, \omega_t) = \left(\rho_t \delta_t(\theta_t) - \phi_{\theta_t}(s_t)^\top \omega_t\right)\nabla^2 V_{\theta_t}(s_t)\omega_t$
  7:     $\omega_{t+1} = \mathbf{\Pi}_{R_\omega}\left(\omega_t + \beta\left(-\phi_{\theta_t}(s_t)\phi_{\theta_t}(s_t)^\top \omega_t + \rho_t \delta_t(\theta_t)\phi_{\theta_t}(s_t)\right)\right)$
  8:     $\theta_{t+1} = \theta_t + \alpha\left(\rho_t\delta_t(\theta_t)\phi_{\theta_t}(s_t) - \gamma\rho_t\phi_{\theta_t}(s_t')\phi_{\theta_t}(s_t)^\top\omega_t - h_t(\theta_t, \omega_t)\right)$
  9: **end for**
**Output**: $\theta_W$
---

We note that under the i.i.d. setting, it is assumed that at each time step $t$, a sample $O_t = (s_t, a_t, r_t, s_t')$ is available, where $s_t \sim \mu^{\pi_b}(\cdot)$, $a_t \sim \pi_b(\cdot|s_t)$ and $s_t' \sim \mathsf{P}(\cdot|s_t, a_t)$.

### B.1  Tracking Error Analysis under the i.i.d. Setting

Denote the tracking error by $z_t = \omega_t - \omega(\theta_t)$. Then by the update of $\omega_t$, the update of $z_t$ can be written as

$$
\begin{aligned}
z_{t+1} &= \omega_{t+1} - \omega(\theta_{t+1})\\
&= \omega_t + \beta\left(-\phi_{\theta_t}(s_t)\phi_{\theta_t}(s_t)^\top\omega_t + \rho_t\delta_t(\theta_t)\phi_{\theta_t}(s_t)\right) - \omega(\theta_{t+1})\\
&= z_t + \omega(\theta_t) - \omega(\theta_{t+1}) + \beta\left(-\phi_{\theta_t}(s_t)\phi_{\theta_t}(s_t)^\top(z_t + \omega(\theta_t)) + \rho_t\delta_t(\theta_t)\phi_{\theta_t}(s_t)\right)\\
&= z_t + \omega(\theta_t) - \omega(\theta_{t+1}) + \beta\left(-A_{\theta_t}(s_t)z_t - A_{\theta_t}(s_t)\omega(\theta_t) + \rho_t\delta_t(\theta_t)\phi_{\theta_t}(s_t)\right), \tag{53}
\end{aligned}
$$

where $A_{\theta_t}(s_t) = \phi_{\theta_t}(s_t)\phi_{\theta_t}(s_t)^\top$. It then follows that

$$
\|z_{t+1}\|^2
$$

$$
\begin{aligned}
&= \|z_t + \omega(\theta_t) - \omega(\theta_{t+1}) + \beta\left(-A_{\theta_t}(s_t)z_t - A_{\theta_t}(s_t)\omega(\theta_t) + \rho_t\delta_t(\theta_t)\phi_{\theta_t}(s_t)\right)\|^2 \\
&= \|z_t\|^2 + \|\omega(\theta_t) - \omega(\theta_{t+1}) + \beta\left(-A_{\theta_t}(s_t)z_t - A_{\theta_t}(s_t)\omega(\theta_t) + \rho_t\delta_t(\theta_t)\phi_{\theta_t}(s_t)\right)\|^2 \\
&\quad + 2\langle z_t, \omega(\theta_t) - \omega(\theta_{t+1})\rangle - 2\beta\langle z_t, A_{\theta_t}(s_t)z_t\rangle + 2\beta\langle z_t, -A_{\theta_t}(s_t)\omega(\theta_t) + \rho_t\delta_t(\theta_t)\phi_{\theta_t}(s_t)\rangle \\
&\leq \|z_t\|^2 + \underbrace{2\beta^2\|\left(-A_{\theta_t}(s_t)z_t - A_{\theta_t}(s_t)\omega(\theta_t) + \rho_t\delta_t(\theta_t)\phi_{\theta_t}(s_t)\right)\|^2}_{(a)} \\
&\quad + \underbrace{2\|\omega(\theta_t) - \omega(\theta_{t+1})\|^2}_{(b)} + \underbrace{2\langle z_t, \omega(\theta_t) - \omega(\theta_{t+1})\rangle}_{(c)} \underbrace{-2\beta\langle z_t, A_{\theta_t}(s_t)z_t\rangle}_{(d)} \\
&\quad + 2\beta\langle z_t, -A_{\theta_t}(s_t)\omega(\theta_t) + \rho_t\delta_t(\theta_t)\phi_{\theta_t}(s_t)\rangle. \tag{54}
\end{aligned}
$$

We then provide the bounds of the terms in (54) one by one. Their proofs can be found in Appendices B.1.1 to B.1.4.

**Term $(a)$ can be bounded as follows:**

$$
2\beta^2\|\left(-A_{\theta_t}(s_t)z_t - A_{\theta_t}(s_t)\omega(\theta_t) + \rho_t\delta_t(\theta_t)\phi_{\theta_t}(s_t)\right)\|^2 \leq 4\beta^2 C_\phi^2\|z_t\|^2 + 4\beta^2 C_{g1}, \tag{55}
$$

where $C_{g1} = \left(\frac{C_\phi^3}{\lambda_v}(r_{\max} + (1+\gamma)C_v) + \rho_{\max}C_\phi(r_{\max} + (1+\gamma)C_v)\right)^2$.

**Term $(b)$ can be bounded as follows:**

$$
2\|\omega(\theta_t) - \omega(\theta_{t+1})\|^2 \leq 4\alpha^2 L_\omega^2 L_g^2\|z_t\|^2 + 4\alpha^2 C_g^2 L_\omega^2, \tag{56}
$$

where $C_g = \rho_{\max}C_\phi(r_{\max} + (1+\gamma)C_v) + \gamma\rho_{\max}R_\omega C_\phi^2 + D_v R_\omega(R_\omega C_\phi + \rho_{\max}(r_{\max} + C_v + \gamma C_v))$.

**Term $(c)$ can be bounded as follows:**

$$
\begin{aligned}
&2\langle z_t, \omega(\theta_t) - \omega(\theta_{t+1})\rangle \\
&\leq 2\left(\alpha L_\omega L_g + \frac{1}{2}\alpha L_\omega + 4\alpha^2 C_g L_g D_\omega\right)\|z_t\|^2 + \frac{\alpha L_\omega}{4}\|\nabla J(\theta_t)\|^2 + \frac{\alpha^2 C_g^3 D_\omega}{L_g} + 2\alpha\eta_G(\theta_t, z_t, O_t),
\end{aligned} \tag{57}
$$

where $\eta_G(\theta_t, z_t, O_t) = -\left\langle z_t, \nabla\omega(\theta_t)\left(G_{t+1}(\theta_t, \omega(\theta_t)) + \frac{\nabla J(\theta_t)}{2}\right)\right\rangle$.

**Term $(d)$ can be bounded as follows:**

$$
-2\beta\langle z_t, A_{\theta_t}(s_t)z_t\rangle \leq -2\beta\lambda_v\|z_t\|^2 + 2\beta\langle z_t, (A_{\theta_t} - A_{\theta_t}(s_t))z_t\rangle, \tag{58}
$$

where $A_\theta = \mathbb{E}_{\mu^{\pi_b}}\left[\phi_\theta(S)\phi_\theta(S)^\top\right]$ is the expectation of $A_\theta(S)$.

By plugging all the bounds from (55), (56), (57) and (58) in (54), it follows that

$$
\begin{aligned}
&\|z_{t+1}\|^2 \\
&\leq \left(1 + 4\beta^2 C_\phi^2 + 4\alpha^2 L_\omega^2 L_g^2 + 2\alpha L_w L_g + \alpha L_w + 8\alpha^2 C_g L_g D_\omega - 2\beta\lambda_v\right)\|z_t\|^2 \\
&\quad + \frac{1}{4}\alpha L_\omega\|\nabla J(\theta_t)\|^2 + 4\beta^2 C_{g1} + 4\alpha^2 C_g^2 L_\omega^2 + \frac{\alpha^2 C_g^3 D_\omega}{L_g} + 2\alpha\eta_G(\theta_t, z_t, O_t) \\
&\quad + 2\beta\langle z_t, (A_{\theta_t} - A_{\theta_t}(s_t))z_t\rangle + 2\beta\langle z_t, -A_{\theta_t}(s_t)\omega(\theta_t) + \rho_t\delta_t(\theta_t)\phi_{\theta_t}(s_t)\rangle \\
&\triangleq (1-q)\|z_t\|^2 + \frac{\alpha L_\omega}{4}\|\nabla J(\theta_t)\|^2 + 4\beta^2 C_{g1} + 4\alpha^2 C_g^2 L_\omega^2 + \frac{\alpha^2 C_g^3 D_\omega}{L_g} + 2\alpha\eta_G(\theta_t, z_t, O_t) \\
&\quad + 2\beta\langle z_t, (A_{\theta_t} - A_{\theta_t}(s_t))z_t\rangle + 2\beta\langle z_t, -A_{\theta_t}(s_t)\omega(\theta_t) + \rho_t\delta_t(\theta_t)\phi_{\theta_t}(s_t)\rangle, \tag{59}
\end{aligned}
$$

where $q = 2\beta\lambda_v - 4\beta^2 C_\phi^2 - 4\alpha^2 L_\omega^2 L_g^2 - 2\alpha L_w L_g - \alpha L_w - 8\alpha^2 C_g L_g D_\omega$. Note that $q = \mathcal{O}(\beta - \beta^2 - \alpha - \alpha^2) = \mathcal{O}(\beta)$, hence we can choose $\alpha$ and $\beta$ such that $q > 0$.

Note that under the i.i.d. setting,

$$
\begin{aligned}
\mathbb{E}\left[\eta_G(\theta_t, z_t, O_t)\right] &= \mathbb{E}\left[\mathbb{E}\left[\eta_G(\theta_t, z_t, O_t)|\mathcal{F}_t\right]\right] \\
&= \mathbb{E}\left[-\left\langle z_t, \nabla\omega(\theta_t)\mathbb{E}\left[\left(G_{t+1}(\theta_t, \omega(\theta_t)) + \frac{\nabla J(\theta_t)}{2}\right)\Big|\mathcal{F}_t\right]\right\rangle\right]
\end{aligned}
$$

$$= 0, \tag{60}$$

which is due to the fact that $\mathbb{E}_{\mu^{\pi_b}}[G_{t+1}(\theta, \omega(\theta))] = -\frac{\nabla J(\theta)}{2}$ when $\theta$ is fixed, and $\mathcal{F}_t$ is the $\sigma$-field generated by the randomness until $\theta_t$ and $\omega_t$. Similarly, it can also be shown that

$$\mathbb{E}[\langle z_t, (A_{\theta_t} - A_{\theta_t}(s_t))z_t \rangle] = 0 \tag{61}$$
$$\mathbb{E}[\langle z_t, -A_{\theta_t}(s_t)\omega(\theta_t) + \rho_t \delta_t(\theta_t)\phi_{\theta_t}(s_t) \rangle] = 0. \tag{62}$$

Hence the tracking error in (59) can be further bounded as

$$\mathbb{E}[\|z_{t+1}\|^2] \le (1-q)\mathbb{E}\left[\|z_t\|^2\right] + \frac{\alpha L_\omega}{4}\mathbb{E}\left[\|\nabla J(\theta_t)\|^2\right] + 4\beta^2 C_{g1} + 4\alpha^2 C_g^2 L_\omega^2 + \frac{\alpha^2 C_g^3 D_\omega}{L_g}. \tag{63}$$

Recursively applying the inequality in (63), it follows that

$$\mathbb{E}\left[\|z_t\|^2\right] \le (1-q)^t \|z_0\|^2 + \frac{\alpha L_\omega}{4}\sum_{i=0}^t (1-q)^{t-i}\mathbb{E}\left[\|\nabla J(\theta_i)\|^2\right]$$
$$+ \frac{1}{q}\left(4\beta^2 C_{g1} + 4\alpha^2 C_g^2 L_\omega^2 + \frac{\alpha^2 C_g^3 D_\omega}{L_g}\right), \tag{64}$$

and summing up w.r.t. $t$ from $0$ to $T-1$, it follows that

$$\frac{\sum_{t=0}^{T-1}\mathbb{E}\left[\|z_t\|^2\right]}{T} \le \frac{\sum_{t=0}^{T-1}(1-q)^t}{T}\|z_0\|^2 + \frac{\alpha L_\omega}{4T}\sum_{t=0}^{T-1}\sum_{i=0}^t (1-q)^{t-i}\mathbb{E}\left[\|\nabla J(\theta_i)\|^2\right]$$
$$+ \frac{1}{q}\left(4\beta^2 C_{g1} + 4\alpha^2 C_g^2 L_\omega^2 + \frac{\alpha^2 C_g^3 D_\omega}{L_g}\right)$$
$$\overset{(a)}{\le} \frac{\|z_0\|^2}{Tq} + \frac{\alpha L_\omega}{4q}\frac{\sum_{t=0}^{T-1}\mathbb{E}\left[\|\nabla J(\theta_t)\|^2\right]}{T}$$
$$+ \frac{1}{q}\left(4\beta^2 C_{g1} + 4\alpha^2 C_g^2 L_\omega^2 + \frac{\alpha^2 C_g^3 D_\omega}{L_g}\right)$$
$$= \mathcal{O}\left(\frac{1}{T\beta} + \frac{\alpha}{\beta}\frac{\sum_{t=0}^{T-1}\mathbb{E}\left[\|\nabla J(\theta_t)\|^2\right]}{T} + \beta\right), \tag{65}$$

where $(a)$ is due to the double-sum trick, i.e., for any $x_i \ge 0$, $\sum_{t=0}^{T-1}\sum_{i=0}^t (1-q)^{t-i}x_i \le \sum_{t=0}^{T-1}(1-q)^t \sum_{t=0}^{T-1} x_t \le \frac{1}{q}\sum_{t=0}^{T-1} x_t$, and the last step is because $q = \mathcal{O}(\beta)$.

### B.1.1 Bound on Term $(a)$

In this section we provide the detailed proof of the bound on term $(a)$ in (55).

It can be shown that

$$\|\left(-A_{\theta_t}(s_t)z_t - A_{\theta_t}(s_t)\omega(\theta_t) + \rho_t \delta_t(\theta_t)\phi_{\theta_t}(s_t)\right)\|^2$$
$$\le 2\|-A_{\theta_t}(s_t)z_t\|^2 + 2\|-A_{\theta_t}(s_t)\omega(\theta_t) + \rho_t \delta_t(\theta_t)\phi_{\theta_t}(s_t)\|^2$$
$$\overset{(a)}{\le} 2C_\phi^2\|z_t\|^2 + 2\left(\frac{C_\phi^3}{\lambda_v}(r_{\max} + (1+\gamma)C_v) + \rho_{\max}C_\phi(r_{\max} + (1+\gamma)C_v)\right)^2, \tag{66}$$

where $(a)$ is from the fact that $\|A_\theta(s)\| = \|\phi_\theta(s)\phi_\theta(s)^\top\| \le C_\phi^2$ and the bounds in (42) and (44).

### B.1.2 Bound on Term $(b)$

In this section we provide the detailed proof of the bound on term $(b)$ in (56).

We first show that $G_{t+1}(\theta, \omega)$ is Lipschitz in $\omega$ for any fixed $\theta$. Specifically, for any $\theta, \omega_1, \omega_2 \in \mathbb{R}^N$, it follows that

$$
\begin{aligned}
&\|G_{t+1}(\theta, \omega_1) - G_{t+1}(\theta, \omega_2)\| \\
&= \|\rho_t \delta_t(\theta) \phi_\theta(s_t) - \gamma \rho_t \phi_\theta(s_t') \phi_\theta(s_t)^\top \omega_1 - h_t(\theta, \omega_1) - \rho_t \delta_t(\theta) \phi_\theta(s_t) + \gamma \rho_t \phi_\theta(s_t') \phi_\theta(s_t)^\top \omega_2 \\
&\quad + h_t(\theta, \omega_2)\| \\
&\leq \|h_t(\theta, \omega_1) - h_t(\theta, \omega_2)\| + \|\gamma \rho_t \phi_\theta(s_t') \phi_\theta(s_t)^\top \omega_1 - \gamma \rho_t \phi_\theta(s_t') \phi_\theta(s_t)^\top \omega_2\| \\
&\overset{(a)}{\leq} \left(C_\phi D_v R_\omega + D_v(C_\phi R_\omega + \rho_{\max}(r_{\max} + C_v + \gamma C_v)) + \gamma \rho_{\max} C_\phi^2\right) \|\omega_1 - \omega_2\| \\
&\triangleq L_g \|\omega_1 - \omega_2\|,
\end{aligned}
\tag{67}
$$

where $L_g = D_v(2C_\phi R_\omega + \rho_{\max}(r_{\max} + C_v + \gamma C_v)) + \gamma \rho_{\max} C_\phi^2$, and $(a)$ is from the Lipschitz continuous of $h_t(\theta, \cdot)$, i.e.,

$$
\|h_t(\theta, \omega_1) - h_t(\theta, \omega_2)\| \leq \rho_{\max}(r_{\max} + (1 + \gamma)C_v)D_v \|\omega_1 - \omega_2\| + 2C_\phi D_v R_\omega \|\omega_1 - \omega_2\|.
\tag{68}
$$

We note that to show (67), we use the bound on $\omega_t$, which is guaranteed by the projection step. And this is the only step in our proof where the projection is used.

Then it follows that

$$
\begin{aligned}
\|\theta_{t+1} - \theta_t\| &= \alpha \|G_{t+1}(\theta_t, \omega_t)\| \\
&\leq \alpha \|G_{t+1}(\theta_t, \omega_t) - G_{t+1}(\theta_t, \omega(\theta_t)) + G_{t+1}(\theta_t, \omega(\theta_t))\| \\
&\leq \alpha L_g \|z_t\| + \alpha \|G_{t+1}(\theta_t, \omega(\theta_t))\| \\
&\leq \alpha L_g \|z_t\| + \alpha C_g,
\end{aligned}
\tag{69}
$$

where $C_g = \rho_{\max} C_\phi(r_{\max} + (1+\gamma)C_v) + \gamma \rho_{\max} R_\omega C_\phi^2 + D_v R_\omega(R_\omega C_\phi + \rho_{\max}(r_{\max} + C_v + \gamma C_v))$, and the last step in (69) can be shown as follows

$$
\begin{aligned}
&\|G_{t+1}(\theta_t, \omega(\theta_t))\| \\
&= \|\rho_t \delta_t(\theta) \phi_\theta(s_t) - \gamma \rho_t \phi_\theta(s_t') \phi_\theta(s_t)^\top \omega(\theta) - h_t(\theta, \omega(\theta))\| \\
&\leq \rho_{\max} C_\phi(r_{\max} + (1 + \gamma)C_v) + \gamma \rho_{\max} R_\omega C_\phi^2 + D_v R_\omega(R_\omega C_\phi + \rho_{\max}(r_{\max} + C_v + \gamma C_v)).
\end{aligned}
\tag{70}
$$

Using (21) and (69), it follows that

$$
\|\omega(\theta_t) - \omega(\theta_{t+1})\| \leq L_\omega \|\theta_{t+1} - \theta_t\| \leq \alpha L_\omega L_g \|z_t\| + \alpha C_g L_\omega,
\tag{71}
$$

and

$$
\|\omega(\theta_t) - \omega(\theta_{t+1})\|^2 \leq 2\alpha^2 L_\omega^2 L_g^2 \|z_t\|^2 + 2\alpha^2 C_g^2 L_\omega^2.
\tag{72}
$$

This completes the proof for term $(b)$.

### B.1.3 Bound on Term $(c)$

In this section we provide the detailed proof of the bound on term $(c)$ in (57).

Consider the inner product $\langle z_t, \omega(\theta_t) - \omega(\theta_{t+1}) \rangle$. By the Mean-Value Theorem, it follows that

$$
\langle z_t, \omega(\theta_t) \rangle - \langle z_t, \omega(\theta_{t+1}) \rangle = \langle z_t, \omega(\theta_t) - \omega(\theta_{t+1}) \rangle = \langle z_t, \nabla \omega(\hat{\theta}_t)(\theta_t - \theta_{t+1}) \rangle,
\tag{73}
$$

where $\hat{\theta}_t = c\theta_t + (1 - c)\theta_{t+1}$ for some $c \in [0, 1]$. Thus, it follows that

$$
\begin{aligned}
&\langle z_t, \omega(\theta_t) - \omega(\theta_{t+1}) \rangle \\
&= \langle z_t, \nabla \omega(\hat{\theta}_t)(\theta_t - \theta_{t+1}) \rangle \\
&= -\alpha \langle z_t, \nabla \omega(\hat{\theta}_t) G_{t+1}(\theta_t, \omega_t) \rangle \\
&= -\alpha \left\langle z_t, \nabla \omega(\hat{\theta}_t) \left( G_{t+1}(\theta_t, \omega_t) - G_{t+1}(\theta_t, \omega(\theta_t)) + G_{t+1}(\theta_t, \omega(\theta_t)) + \frac{\nabla J(\theta_t)}{2} \right) \right\rangle
\end{aligned}
$$

$$+ \alpha \left\langle z_t, \nabla \omega(\hat{\theta}_t) \frac{\nabla J(\theta_t)}{2} \right\rangle$$

$$= -\alpha \left\langle z_t, \nabla \omega(\hat{\theta}_t) \left( G_{t+1}(\theta_t, \omega_t) - G_{t+1}(\theta_t, \omega(\theta_t)) \right) \right\rangle + \alpha \left\langle z_t, \nabla \omega(\hat{\theta}_t) \frac{\nabla J(\theta_t)}{2} \right\rangle$$

$$- \alpha \left\langle z_t, \nabla \omega(\hat{\theta}_t) \left( G_{t+1}(\theta_t, \omega(\theta_t)) + \frac{\nabla J(\theta_t)}{2} \right) \right\rangle$$

$$\overset{(a)}{\leq} \alpha L_\omega L_g \|z_t\|^2 + \alpha L_\omega \|z_t\| \left\| \frac{\nabla J(\theta_t)}{2} \right\| - \alpha \left\langle z_t, \nabla \omega(\theta_t) \left( G_{t+1}(\theta_t, \omega(\theta_t)) + \frac{\nabla J(\theta_t)}{2} \right) \right\rangle$$

$$+ \alpha \left\langle z_t, (\nabla \omega(\theta_t) - \nabla \omega(\hat{\theta}_t)) \left( G_{t+1}(\theta_t, \omega(\theta_t)) + \frac{\nabla J(\theta_t)}{2} \right) \right\rangle$$

$$\leq \alpha L_\omega L_g \|z_t\|^2 + \frac{1}{2} \alpha L_\omega \|z_t\|^2 + \frac{\alpha L_\omega}{8} \|\nabla J(\theta_t)\|^2 + \alpha \eta_G(\theta_t, z_t, O_t)$$

$$+ \alpha \|z_t\| \|\nabla \omega(\theta_t) - \nabla \omega(\hat{\theta}_t)\| \left\| G_{t+1}(\theta_t, \omega(\theta_t)) + \frac{\nabla J(\theta_t)}{2} \right\|$$

$$\overset{(b)}{\leq} \alpha L_\omega L_g \|z_t\|^2 + \frac{1}{2} \alpha L_\omega \|z_t\|^2 + \frac{\alpha L_\omega}{8} \|\nabla J(\theta_t)\|^2 + \alpha \eta_G(\theta_t, z_t, O_t) + 2\alpha C_g D_\omega \|z_t\| \|\theta_t - \hat{\theta}_t\|$$

$$\overset{(c)}{\leq} \alpha L_\omega L_g \|z_t\|^2 + \frac{1}{2} \alpha L_\omega \|z_t\|^2 + \frac{\alpha L_\omega}{8} \|\nabla J(\theta_t)\|^2 + \alpha \eta_G(\theta_t, z_t, O_t)$$

$$+ 2\alpha C_g D_\omega \|z_t\| \|\theta_t - \theta_{t+1}\|$$

$$\overset{(d)}{\leq} \alpha L_\omega L_g \|z_t\|^2 + \frac{1}{2} \alpha L_\omega \|z_t\|^2 + \frac{\alpha L_\omega}{8} \|\nabla J(\theta_t)\|^2 + \alpha \eta_G(\theta_t, z_t, O_t)$$

$$+ 2\alpha C_g D_\omega \|z_t\| (\alpha L_g \|z_t\| + \alpha C_g)$$

$$\overset{(e)}{\leq} \alpha L_\omega L_g \|z_t\|^2 + \frac{1}{2} \alpha L_\omega \|z_t\|^2 + \frac{\alpha L_\omega}{8} \|\nabla J(\theta_t)\|^2 + \alpha \eta_G(\theta_t, z_t, O_t)$$

$$+ 2\alpha^2 C_g D_\omega \left( 2L_g \|z_t\|^2 + \frac{C_g^2}{4L_g} \right)$$

$$\leq (\alpha L_\omega L_g + \frac{1}{2} \alpha L_\omega + 4\alpha^2 C_g L_g D_\omega) \|z_t\|^2 + \frac{\alpha L_\omega}{8} \|\nabla J(\theta_t)\|^2 + \frac{\alpha^2 C_g^3 D_\omega}{2L_g} + \alpha \eta_G(\theta_t, z_t, O_t),$$
$$\tag{74}$$

where $\eta_G(\theta_t, z_t, O_t) = -\left\langle z_t, \nabla \omega(\theta_t) \left( G_{t+1}(\theta_t, \omega(\theta_t)) + \frac{\nabla J(\theta_t)}{2} \right) \right\rangle$, $(a)$ is from the Lipschitz continuity of $G_{t+1}(\theta, \cdot)$ proved in (67), $(b)$ is from the Lipschitz continuity of $\nabla \omega(\theta)$, which is shown in (30), $(c)$ is from the fact that $\|\theta_t - \hat{\theta}_t\| = (1-c)\|\theta_t - \theta_{t+1}\| \leq \|\theta_t - \theta_{t+1}\|$, $(d)$ is from the bound of $\|\theta_t - \theta_{t+1}\|$ in (69), and $(e)$ is from the fact that $C_g \|z_t\| \leq L_g \|z_t\|^2 + \frac{C_g^2}{4L_g}$.

This completes the proof.

### B.1.4 Bound on Term $(d)$

In this section we provide the detailed proof of the bound on term $(d)$ in (58).

It can be shown that

$$-2\beta \langle z_t, A_{\theta_t}(s_t) z_t \rangle = -2\beta \langle z_t, A_{\theta_t} z_t \rangle + 2\beta \langle z_t, (A_{\theta_t} - A_{\theta_t}(s_t)) z_t \rangle$$
$$\leq -2\beta \lambda_v \|z_t\|^2 + 2\beta \langle z_t, (A_{\theta_t} - A_{\theta_t}(s_t)) z_t \rangle, \tag{75}$$

where the inequality is due to the fact that $\langle z_t, A_{\theta_t} z_t \rangle = z_t^\top A_{\theta_t} z_t \geq \lambda_L(A_{\theta_t}) \|z_t\|^2 \geq \lambda_v \|z_t\|^2$.

## B.2 Proof under the i.i.d. Setting

In this section we provide the proof of Theorem 1 under the i.i.d. setting.

From Lemma 3, we know that the objective function $J(\theta)$ is $L_J$-smooth, hence it follows that

$$
\begin{aligned}
J(\theta_{t+1}) &\leq J(\theta_t) + \langle \nabla J(\theta_t), \theta_{t+1} - \theta_t \rangle + \frac{L_J}{2}\|\theta_{t+1} - \theta_t\|^2 \\
&= J(\theta_t) + \alpha \langle \nabla J(\theta_t), G_{t+1}(\theta_t, \omega_t) \rangle + \frac{L_J}{2}\alpha^2\|G_{t+1}(\theta_t, \omega_t)\|^2 \\
&= J(\theta_t) - \alpha \left\langle \nabla J(\theta_t), -G_{t+1}(\theta_t, \omega_t) - \frac{\nabla J(\theta_t)}{2} + G_{t+1}(\theta_t, \omega(\theta_t)) - G_{t+1}(\theta_t, \omega(\theta_t)) \right\rangle \\
&\quad - \frac{\alpha}{2}\|\nabla J(\theta_t)\|^2 + \frac{L_J}{2}\alpha^2\|G_{t+1}(\theta_t, \omega_t)\|^2 \\
&= J(\theta_t) - \alpha \langle \nabla J(\theta_t), -G_{t+1}(\theta_t, \omega_t) + G_{t+1}(\theta_t, \omega(\theta_t)) \rangle \\
&\quad + \alpha \left\langle \nabla J(\theta_t), \frac{\nabla J(\theta_t)}{2} + G_{t+1}(\theta_t, \omega(\theta_t)) \right\rangle - \frac{\alpha}{2}\|\nabla J(\theta_t)\|^2 + \frac{L_J}{2}\alpha^2\|G_{t+1}(\theta_t, \omega_t)\|^2 \\
&\overset{(a)}{\leq} J(\theta_t) + \alpha L_g\|\nabla J(\theta_t)\|\|\omega(\theta_t) - \omega_t\| + \alpha \left\langle \nabla J(\theta_t), \frac{\nabla J(\theta_t)}{2} + G_{t+1}(\theta_t, \omega(\theta_t)) \right\rangle \\
&\quad - \frac{\alpha}{2}\|\nabla J(\theta_t)\|^2 + \frac{L_J}{2}\alpha^2\|G_{t+1}(\theta_t, \omega_t)\|^2 \\
&\overset{(b)}{\leq} J(\theta_t) + \alpha L_g\|\nabla J(\theta_t)\|\|z_t\| + \alpha \left\langle \nabla J(\theta_t), \frac{\nabla J(\theta_t)}{2} + G_{t+1}(\theta_t, \omega(\theta_t)) \right\rangle \\
&\quad - \frac{\alpha}{2}\|\nabla J(\theta_t)\|^2 + \frac{L_J}{2}\alpha^2 \left(2L_g^2\|z_t\|^2 + 2C_g^2\right),
\end{aligned}
\tag{76}
$$

where $(a)$ is from (67) and $(b)$ is because $\|\theta_{t+1} - \theta_t\| = \alpha\|G_{t+1}(\theta_t, \omega_t)\| \leq \alpha L_g\|z_t\| + \alpha C_g$, whose detailed proof is provided in (69). Thus by re-arranging the terms, taking expectation and summing up w.r.t. $t$ from 0 to $T-1$, it follows that

$$
\begin{aligned}
&\frac{\alpha}{2}\sum_{t=0}^{T-1}\mathbb{E}[\|\nabla J(\theta_t)\|^2] \\
&\leq -\mathbb{E}[J(\theta_T)] + J(\theta_0) + \alpha L_g\sqrt{\sum_{t=0}^{T-1}\mathbb{E}[\|\nabla J(\theta_t)\|^2]}\sqrt{\sum_{t=0}^{T-1}\mathbb{E}[\|z_t\|^2]} + \alpha^2 L_J L_g^2\sum_{t=0}^{T-1}\mathbb{E}[\|z_t\|^2] \\
&\quad + \alpha^2 C_g^2 L_J T,
\end{aligned}
\tag{77}
$$

which is due to the fact that under the i.i.d. setting,

$$
\begin{aligned}
&\mathbb{E}\left[\left\langle \nabla J(\theta_t), \frac{\nabla J(\theta_t)}{2} + G_{t+1}(\theta_t, \omega(\theta_t)) \right\rangle\right] \\
&= \mathbb{E}\left[\left\langle \nabla J(\theta_t), \mathbb{E}\left[\frac{\nabla J(\theta_t)}{2} + G_{t+1}(\theta_t, \omega(\theta_t))\Big|\mathcal{F}_t\right]\right\rangle\right] = 0,
\end{aligned}
\tag{78}
$$

and the Cauchy's inequality

$$
\sum_{t=0}^{T-1}\mathbb{E}[\|\nabla J(\theta_t)\|\|z_t\|] \leq \sqrt{\sum_{t=0}^{T-1}\mathbb{E}[\|\nabla J(\theta_t)\|^2]}\sqrt{\sum_{t=0}^{T-1}\mathbb{E}[\|z_t\|^2]}.
\tag{79}
$$

Thus dividing both sides by $\frac{\alpha T}{2}$, it follows that

$$
\begin{aligned}
&\frac{\sum_{t=0}^{T-1}\mathbb{E}[\|\nabla J(\theta_t)\|^2]}{T} \\
&\leq \frac{2(J(\theta_0) - J^*)}{T\alpha} + 2L_g\sqrt{\frac{\sum_{t=0}^{T-1}\mathbb{E}[\|\nabla J(\theta_t)\|^2]}{T}}\sqrt{\frac{\sum_{t=0}^{T-1}\mathbb{E}[\|z_t\|^2]}{T}} \\
&\quad + 2\alpha L_J L_g^2\frac{\sum_{t=0}^{T-1}\mathbb{E}[\|z_t\|^2]}{T} + 2\alpha C_g^2 L_J,
\end{aligned}
\tag{80}
$$

where $J^* \triangleq \min_\theta J(\theta)$.

Recall the tracking error in (65):

$$\frac{\sum_{t=0}^{T-1} \mathbb{E}\left[\|z_t\|^2\right]}{T}$$

$$\leq \frac{\|z_0\|^2}{Tq} + \frac{\alpha L_\omega}{4q} \frac{\sum_{t=0}^{T-1} \mathbb{E}\left[\|\nabla J(\theta_t)\|^2\right]}{T} + \frac{1}{q}\left(4\beta^2 C_{g1} + 4\alpha^2 C_g^2 L_\omega^2 + \frac{\alpha^2 C_g^3 D_\omega}{L_g}\right). \qquad (81)$$

We then plug in the tracking error and obtain that

$$\frac{\sum_{t=0}^{T-1} \mathbb{E}[\|\nabla J(\theta_t)\|^2]}{T}$$

$$\leq \frac{2(J(\theta_0) - J^*)}{T\alpha} + 2\alpha C_g^2 L_J + 2L_g \sqrt{\frac{\sum_{t=0}^{T-1} \mathbb{E}[\|\nabla J(\theta_t)\|^2]}{T}}$$

$$\times \sqrt{\frac{\|z_0\|^2}{Tq} + \alpha L_\omega \frac{1}{4q} \frac{\sum_{t=0}^{T-1} \mathbb{E}\left[\|\nabla J(\theta_t)\|^2\right]}{T} + \frac{1}{q}\left(4\beta^2 C_{g1} + 4\alpha^2 C_g^2 L_\omega^2 + \frac{\alpha^2 C_g^3 D_\omega}{L_g}\right)}$$

$$+ 2\alpha L_J L_g^2 \left(\frac{\|z_0\|^2}{Tq} + \alpha L_\omega \frac{1}{4q} \frac{\sum_{t=0}^{T-1} \mathbb{E}\left[\|\nabla J(\theta_t)\|^2\right]}{T}\right.$$

$$\left. + \frac{1}{q}\left(4\beta^2 C_{g1} + 4\alpha^2 C_g^2 L_\omega^2 + \frac{\alpha^2 C_g^3 D_\omega}{L_g}\right)\right)$$

$$\leq \frac{2(J(\theta_0) - J^*)}{T\alpha} + 2\alpha C_g^2 L_J + L_g \sqrt{\frac{\alpha L_\omega}{q} \frac{\sum_{t=0}^{T-1} \mathbb{E}\left[\|\nabla J(\theta_t)\|^2\right]}{T}}$$

$$+ 2L_g \sqrt{\frac{\sum_{t=0}^{T-1} \mathbb{E}[\|\nabla J(\theta_t)\|^2]}{T}} \sqrt{\frac{\|z_0\|^2}{Tq} + \frac{1}{q}\left(4\beta^2 C_{g1} + 4\alpha^2 C_g^2 L_\omega^2 + \frac{\alpha^2 C_g^3 D_\omega}{L_g}\right)}$$

$$+ 2\alpha L_J L_g^2 \left(\frac{\|z_0\|^2}{Tq} + \alpha L_\omega \frac{1}{4q} \frac{\sum_{t=0}^{T-1} \mathbb{E}\left[\|\nabla J(\theta_t)\|^2\right]}{T}\right.$$

$$\left. + \frac{1}{q}\left(4\beta^2 C_{g1} + 4\alpha^2 C_g^2 L_\omega^2 + \frac{\alpha^2 C_g^3 D_\omega}{L_g}\right)\right), \qquad (82)$$

where the last step is from the fact that $\sqrt{x+y} \leq \sqrt{x} + \sqrt{y}$ for any $x, y \geq 0$. Re-arranging the terms, it follows that

$$\left(1 - L_g\sqrt{\frac{\alpha L_\omega}{q}} - \frac{\alpha^2 L_J L_g^2 L_\omega}{2q}\right) \frac{\sum_{t=0}^{T-1} \mathbb{E}[\|\nabla J(\theta_t)\|^2]}{T}$$

$$\leq \frac{2(J(\theta_0) - J^*)}{T\alpha} + 2\alpha C_g^2 L_J + 2\alpha L_J L_g^2 \left(\frac{\|z_0\|^2}{Tq} + \frac{1}{q}\left(4\beta^2 C_{g1} + 4\alpha^2 C_g^2 L_\omega^2 + \frac{\alpha^2 C_g^3 D_\omega}{L_g}\right)\right)$$

$$+ 2L_g \sqrt{\frac{\sum_{t=0}^{T-1} \mathbb{E}[\|\nabla J(\theta_t)\|^2]}{T}} \sqrt{\frac{\|z_0\|^2}{Tq} + \frac{1}{q}\left(4\beta^2 C_{g1} + 4\alpha^2 C_g^2 L_\omega^2 + \frac{\alpha^2 C_g^3 D_\omega}{L_g}\right)}. \qquad (83)$$

Note that $\left(L_g\sqrt{\frac{\alpha L_\omega}{q}} + \frac{\alpha^2 L_J L_g^2 L_\omega}{2q}\right) = \mathcal{O}\left(\sqrt{\frac{\alpha}{\beta}} + \frac{\alpha^2}{\beta}\right)$, hence we can choose $\alpha$ and $\beta$ such that $\left(1 - L_g\sqrt{\frac{\alpha L_\omega}{q}} - \frac{\alpha^2 L_J L_g^2 L_\omega}{2q}\right) \geq \frac{1}{2}$. Thus (83) implies that

$$\frac{\sum_{t=0}^{T-1} \mathbb{E}[\|\nabla J(\theta_t)\|^2]}{T}$$

$$\leq \frac{4(J(\theta_0) - J^*)}{T\alpha} + 4\alpha C_g^2 L_J + 4\alpha L_J L_g^2 \left(\frac{\|z_0\|^2}{Tq} + \frac{1}{q}\left(4\beta^2 C_{g1} + 4\alpha^2 C_g^2 L_\omega^2 + \frac{\alpha^2 C_g^3 D_\omega}{L_g}\right)\right)$$

$$+ 4L_g \sqrt{\frac{\sum_{t=0}^{T-1} \mathbb{E}[\|\nabla J(\theta_t)\|^2]}{T}} \sqrt{\frac{\|z_0\|^2}{Tq} + \frac{1}{q}\left(4\beta^2 C_{g1} + 4\alpha^2 C_g^2 L_\omega^2 + \frac{\alpha^2 C_g^3 D_\omega}{L_g}\right)}. \tag{84}$$

Denote $U = \frac{4(J(\theta_0) - J^*)}{T\alpha} + 4\alpha C_g^2 L_J + 4\alpha L_J L_g^2 \left(\frac{\|z_0\|^2}{Tq} + \frac{1}{q}\left(4\beta^2 C_{g1} + 4\alpha^2 C_g^2 L_\omega^2 + \frac{\alpha^2 C_g^3 D_\omega}{L_g}\right)\right)$,

and $V = 4L_g \sqrt{\frac{\|z_0\|^2}{Tq} + \frac{1}{q}\left(4\beta^2 C_{g1} + 4\alpha^2 C_g^2 L_\omega^2 + \frac{\alpha^2 C_g^3 D_\omega}{L_g}\right)}$. Then it follows that

$$\frac{\sum_{t=0}^{T-1} \mathbb{E}[\|\nabla J(\theta_t)\|^2]}{T} \leq V \sqrt{\frac{\sum_{t=0}^{T-1} \mathbb{E}[\|\nabla J(\theta_t)\|^2]}{T}} + U, \tag{85}$$

which further implies that

$$\frac{\sum_{t=0}^{T-1} \mathbb{E}[\|\nabla J(\theta_t)\|^2]}{T}$$

$$\leq V^2 + 2U$$

$$= 16L_g^2 \left(\frac{\|z_0\|^2}{Tq} + \frac{1}{q}\left(4\beta^2 C_{g1} + 4\alpha^2 C_g^2 L_\omega^2 + \frac{\alpha^2 C_g^3 D_\omega}{L_g}\right)\right) + \frac{8(J(\theta_0) - J^*)}{T\alpha}$$

$$\quad + 8\alpha C_g^2 L_J + 8\alpha L_J L_g^2 \left(\frac{\|z_0\|^2}{Tq} + \frac{1}{q}\left(4\beta^2 C_{g1} + 4\alpha^2 C_g^2 L_\omega^2 + \frac{\alpha^2 C_g^3 D_\omega}{L_g}\right)\right)$$

$$= (16L_g^2 + 8\alpha L_J L_g^2)\left(\frac{\|z_0\|^2}{Tq} + \frac{1}{q}\left(4\beta^2 C_{g1} + 4\alpha^2 C_g^2 L_\omega^2 + \frac{\alpha^2 C_g^3 D_\omega}{L_g}\right)\right)$$

$$\quad + \frac{8(J(\theta_0) - J^*)}{T\alpha} + 8\alpha C_g^2 L_J$$

$$= \mathcal{O}\left(\frac{1}{T\beta} + \beta + \frac{1}{T\alpha}\right)$$

$$= \mathcal{O}\left(\frac{1}{T^{1-a}} + \frac{1}{T^b} + \frac{1}{T^{1-b}}\right). \tag{86}$$

This completes the proof.

### B.3 Choice of Step-sizes

As the proof is complicated and we have made several assumptions on the step-sizes, in this section we summarize all the assumptions we made on the step-sizes. This would help the readers to have a more clear understanding of the choice of $\alpha$ and $\beta$.

In the proof under the i.i.d. setting, we made two assumptions on step-sizes. In (59), we assume
$$q = 2\beta\lambda_v - 4\beta^2 C_\phi^2 - 4\alpha^2 L_\omega^2 L_g^2 - 2\alpha L_w L_g - \alpha L_w - 8\alpha^2 C_g L_g D_\omega > 0; \tag{87}$$
And in (83), we moreover assume
$$\left(1 - L_g \sqrt{\frac{\alpha L_\omega}{q}} - \frac{\alpha^2 L_J L_g^2 L_\omega}{2q}\right) \geq \frac{1}{2}. \tag{88}$$

Note that the first one can be satisfied if $\beta \leq \min\left\{1, \frac{\lambda_v}{4C_\phi^2}\right\}$ and $\frac{\alpha}{\beta} \leq \frac{\lambda_v}{4L_\omega^2 L_g^2 + 2L_w L_g + L_w + 8C_g L_g D_\omega}$. As for assumption (88), we only need to find $\alpha$ and $\beta$ such that

$$L_g \sqrt{\frac{\alpha L_\omega}{q}} \leq \frac{1}{4},$$
$$\frac{\alpha^2 L_J L_g^2 L_\omega}{2q} \leq \frac{1}{4}. \tag{89}$$

Note that these two conditions are satisfied if condition (87) is satisfied.

Hence to meet all the requirements on the step-sizes, we can set $\beta \leq \min\left\{1, \frac{\lambda_v}{4C_\phi^2}\right\}$ and $\frac{\alpha}{\beta} \leq \min\left\{1, \frac{\lambda_v}{4L_\omega^2 L_g^2 + 2L_w L_g + L_w + 8C_g L_g D_\omega}\right\}$.

# C Non-asymptotic Analysis under the Markovian Setting

In this section we provide the proof of Theorem 1 under that Markovian setting. In Appendix C.1 we develop the finite-time analysis of the tracking error and in Appendix C.2 we prove Theorem 1.

## C.1 Tracking Error Analysis under the Markovian Setting

We first define the mixing time $\tau_\beta = \inf \{t : m\kappa^t \leq \beta\}$ (Assumption 4). It can be shown that for any bounded function $\|f(O_t)\| \leq C_f$, for any $t \geq \tau_\beta$, $\|\mathbb{E}[f(O_t)] - \mathbb{E}_{O \sim \mu^{\pi_b}}[f(O)]\| \leq C_f \beta$ and $\tau_\beta = \mathcal{O}(-\log \beta)$. We note that $\beta\tau_\beta \to 0$ as $\beta \to 0$, and we assume that $\beta\tau_\beta C_\phi^2 \leq \frac{1}{4}$.

From (53), the update of the tracking error $z_t$ can be written as

$$z_{t+1} = z_t + \beta(-A_{\theta_t}(s_t)z_t + b_t(\theta_t)) + \omega(\theta_t) - \omega(\theta_{t+1}), \tag{90}$$

where $A_{\theta_t}(s_t) = \phi_{\theta_t}(s_t)\phi_{\theta_t}(s_t)^\top$ and $b_t(\theta_t) = -A_{\theta_t}(s_t)\omega(\theta_t) + \rho_t\delta_t(\theta_t)\phi_{\theta_t}(s_t)$. Note that for any $\theta \in \mathbb{R}^N$ and any sample $O_t = (s_t, a_t, r_t, s_{t+1}) \in \mathcal{S} \times \mathcal{A} \times \mathbb{R} \times \mathcal{S}$, $\|b_t(\theta_t)\| \leq C_\phi^2 R_\omega + \rho_{\max}C_\phi(r_{\max} + C_v + \gamma C_v) \triangleq b_{\max}$.

Then it can be shown that

$$\mathbb{E}\left[\|z_{t+1}\|^2 - \|z_t\|^2\right]$$
$$= \mathbb{E}\left[2z_t^\top(z_{t+1} - z_t) + \|z_{t+1} - z_t\|^2\right]$$
$$= \mathbb{E}\left[2z_t^\top(z_{t+1} - z_t + \beta A_{\theta_t}z_t)\right] + \mathbb{E}\left[\|z_{t+1} - z_t\|^2\right] + \beta\mathbb{E}\left[2z_t^\top(-A_{\theta_t})z_t\right]$$
$$\leq \underbrace{\mathbb{E}\left[\|z_{t+1} - z_t\|^2\right]}_{(a)} + \underbrace{\mathbb{E}\left[2z_t^\top(z_{t+1} - z_t + \beta A_{\theta_t}z_t)\right]}_{(b)} - 2\beta\lambda_v\mathbb{E}\left[\|z_t\|^2\right], \tag{91}$$

where the last inequality is due to the fact that $\lambda_L(A_{\theta_t}) \geq \lambda_v$. We first provide the bounds on terms $(a)$ and $(b)$ as follows, and their detailed proof can be found in Appendices C.1.1 and C.1.2.

**Term $(a)$ can be bounded as follows:**

For any $t \geq 0$, we have that

$$\|z_{t+1} - z_t\|^2 \leq 2\beta^2 C_\phi^4\|z_t\|^2 + 2\beta^2(b_{\max} + L_\omega C_g)^2. \tag{92}$$

**Term $(b)$ can be bounded as follows:**

For any $t \geq \tau_\beta$, we have that

$$\left|\mathbb{E}\left[z_t^\top\left(-A_{\theta_t}z_t - \frac{1}{\beta}(z_{t+1} - z_t)\right)\right]\right|$$
$$\leq (R_1 + R_3 + P_1 + P_2 + P_3)\mathbb{E}\left[\|z_t\|^2\right] + (Q_1 + Q_2 + Q_3 + P_1 + P_2 + P_3)$$
$$+ \frac{\alpha}{8\beta}L_\omega\mathbb{E}\left[\|\nabla J(\theta_t)\|^2\right], \tag{93}$$

where the definition of $P_i, Q_i$ and $R_i$, $i = 1, 2, 3$, can be found in (114), (117) and (120).

From (91), it can be shown that for any $t \geq \tau_\beta$,

$$\mathbb{E}\left[\|z_{t+1}\|^2 - \|z_t\|^2\right]$$
$$\leq 2\beta(R_1 + R_3 + P_1 + P_2 + P_3)\mathbb{E}\left[\|z_t\|^2\right] + 2\beta(Q_1 + Q_2 + Q_3 + P_1 + P_2 + P_3)$$
$$+ \frac{\alpha}{4}L_\omega\mathbb{E}\left[\|\nabla J(\theta_t)\|^2\right] + 2\beta^2 C_\phi^4\mathbb{E}\left[\|z_t\|^2\right] + 2\beta^2(b_{\max} + L_\omega C_g)^2 - 2\beta\lambda_v\mathbb{E}\left[\|z_t\|^2\right]. \tag{94}$$

Thus by re-arranging the terms we obtain that

$$\mathbb{E}\left[\|z_{t+1}\|^2\right]$$
$$\leq (1 - 2\beta\lambda_v + 2\beta(R_1 + R_3 + P_1 + P_2 + P_3) + 2\beta^2 C_\phi^4)\mathbb{E}\left[\|z_t\|^2\right] + \frac{\alpha}{4}L_\omega\mathbb{E}\left[\|\nabla J(\theta_t)\|^2\right]$$

$$+ 2\beta(Q_1 + Q_2 + Q_3 + P_1 + P_2 + P_3) + 2\beta^2(b_{\max} + L_\omega C_g)^2$$

$$\triangleq (1 - q)\mathbb{E}\left[\|z_t\|^2\right] + \frac{\alpha}{4}L_\omega \mathbb{E}\left[\|\nabla J(\theta_t)\|^2\right] + p, \tag{95}$$

where $q = 2\beta\lambda_v - 2\beta(R_1 + R_3 + P_1 + P_2 + P_3) - 2\beta^2 C_\phi^4 = \mathcal{O}(\beta)$ and $p = 2\beta(Q_1 + Q_2 + Q_3 + P_1 + P_2 + P_3) + 2\beta^2(b_{\max} + L_\omega C_g)^2 = \mathcal{O}(\beta^2\tau_\beta)$. Then by recursively using the previous inequality, it follows that for any $t \geq \tau_\beta$,

$$\mathbb{E}[\|z_t\|^2] \leq (1 - q)^{t - \tau_\beta}\mathbb{E}\left[\|z_{\tau_\beta}\|^2\right] + \frac{\alpha L_\omega}{4}\sum_{j=0}^{t}(1 - q)^{t-j}\mathbb{E}[\|\nabla J(\theta_j)\|^2] + \frac{p}{q}, \tag{96}$$

and hence

$$\frac{\sum_{t=0}^{T-1}\mathbb{E}[\|z_t\|^2]}{T}$$

$$= \frac{\sum_{t=\tau_\beta}^{T-1}\mathbb{E}[\|z_t\|^2]}{T} + \frac{\sum_{t=0}^{\tau_\beta-1}\mathbb{E}[\|z_t\|^2]}{T}$$

$$\leq \frac{\mathbb{E}\left[\|z_{\tau_\beta}\|^2\right]}{Tq} + \frac{\tau_\beta\left(2\|z_0\| + 2\beta\tau_\beta(b_{\max} + L_\omega C_g)\right)^2}{T} + \frac{\alpha L_\omega}{4q}\frac{\sum_{t=0}^{T-1}\mathbb{E}[\|\nabla J(\theta_t)\|^2]}{T} + \frac{p}{q}$$

$$\leq \left(2\|z_0\| + 2\beta\tau_\beta(b_{\max} + L_\omega C_g)\right)^2\left(\frac{1}{Tq} + \frac{\tau_\beta}{T}\right) + \frac{\alpha L_\omega}{4q}\frac{\sum_{t=0}^{T-1}\mathbb{E}[\|\nabla J(\theta_t)\|^2]}{T} + \frac{p}{q}$$

$$= \mathcal{O}\left(\frac{1}{T\beta} + \frac{\alpha}{\beta}\frac{\sum_{t=0}^{T-1}\mathbb{E}[\|\nabla J(\theta_t)\|^2]}{T} + \beta\tau_\beta\right), \tag{97}$$

where the last step is because $q = \mathcal{O}(\beta)$ and $p = \mathcal{O}(\beta^2\tau_\beta)$.

### C.1.1  Bound on Term $(a)$

In this section we provide the detailed proof of the bound on term $(a)$ in (91).

We first note that from the update of $z_t$ in (90), term $\|z_{t+1} - z_t\|$ can be bounded as follows

$$\|z_{t+1} - z_t\| \leq \|\beta(-A_{\theta_t}(s_t)z_t + b_t(\theta_t))\| + \|\omega(\theta_t) - \omega(\theta_{t+1})\|$$

$$\leq \beta C_\phi^2\|z_t\| + \beta b_{\max} + L_\omega\|\theta_t - \theta_{t+1}\|$$

$$\overset{(a)}{\leq} \beta C_\phi^2\|z_t\| + \beta b_{\max} + \alpha L_\omega C_g$$

$$\leq \beta C_\phi^2\|z_t\| + \beta(b_{\max} + L_\omega C_g), \tag{98}$$

where $(a)$ is due to the fact $\|G_{t+1}(\theta_t, \omega_t)\| \leq C_g$ for any $t \geq 0$, and where the last inequality is from the fact that $\alpha \leq \beta$. Hence term $(a)$ can be bounded as follows

$$\|z_{t+1} - z_t\|^2 \leq 2\beta^2 C_\phi^4\|z_t\|^2 + 2\beta^2(b_{\max} + L_\omega C_g)^2. \tag{99}$$

This completes the proof.

### C.1.2  Bound on Term $(b)$

In this section we provide the detailed proof of the bound on term $(b)$ in (91).

From (98), it follows that

$$\|z_{t+1}\| \leq (1 + \beta C_\phi^2)\|z_t\| + \beta b_{\max} + \alpha L_\omega C_g$$

$$\leq (1 + \beta C_\phi^2)\|z_t\| + \beta(b_{\max} + L_\omega C_g). \tag{100}$$

By applying (100) recursively, it follows that

$$\|z_t\| \leq (1 + \beta C_\phi^2)^t\|z_0\| + \beta(b_{\max} + L_\omega C_g)\frac{(1 + \beta C_\phi^2)^t - 1}{\beta C_\phi^2}$$

$$= (1 + \beta C_\phi^2)^t \|z_0\| + (b_{\max} + L_\omega C_g) \frac{(1 + \beta C_\phi^2)^t - 1}{C_\phi^2}. \tag{101}$$

We first show the following lemma which bounds the update $\|z_t - z_{t-\tau_\beta}\|$ by $\|z_t\|$.

**Lemma 4.** *For any $t \geq \tau_\beta$ and $t \geq j \geq t - \tau_\beta$, we have that*

$$\|z_j\| \leq 2\|z_{t-\tau_\beta}\| + 2\beta\tau_\beta(b_{\max} + L_\omega C_g); \tag{102}$$

$$\|z_t - z_{t-\tau_\beta}\| \leq 2\beta\tau_\beta C_\phi^2\|z_{t-\tau_\beta}\| + 2\beta\tau_\beta(b_{\max} + L_\omega C_g), \tag{103}$$

$$\|z_t - z_{t-\tau_\beta}\| \leq 4\beta\tau_\beta C_\phi^2\|z_t\| + 4\beta\tau_\beta(b_{\max} + L_\omega C_g). \tag{104}$$

*Proof.* From (100), it follows that

$$\|z_{t+1}\| \leq (1 + \beta C_\phi^2)\|z_t\| + \beta(b_{\max} + L_\omega C_g). \tag{105}$$

First note that $\beta C_\phi^2 \tau_\beta \leq \frac{1}{4}$ and hence $\beta C_\phi^2 \leq \frac{1}{4\tau_\beta} \leq \frac{\log 2}{\tau_\beta - 1}$. This implies that

$$(1 + \beta C_\phi^2)^{\tau_\beta} \leq 1 + 2\tau_\beta \beta C_\phi^2, \tag{106}$$

which is because $(1 + x)^k \leq 1 + 2kx$ for $x \leq \frac{\log 2}{k-1}$.

Applying inequality (105) recursively, it follows that

$$\|z_j\| \leq (1 + \beta C_\phi^2)^{j-t+\tau_\beta}\|z_{t-\tau_\beta}\| + (b_{\max} + L_\omega C_g)\frac{(1 + \beta C_\phi^2)^{\tau_\beta} - 1}{C_\phi^2}$$

$$\leq (1 + \beta C_\phi^2)^{\tau_\beta}\|z_{t-\tau_\beta}\| + (b_{\max} + L_\omega C_g)\frac{(1 + \beta C_\phi^2)^{\tau_\beta} - 1}{C_\phi^2}$$

$$\overset{(a)}{\leq} (1 + 2\tau_\beta \beta C_\phi^2)\|z_{t-\tau_\beta}\| + 2\beta\tau_\beta(b_{\max} + L_\omega C_g)$$

$$\overset{(b)}{\leq} 2\|z_{t-\tau_\beta}\| + 2\beta\tau_\beta(b_{\max} + L_\omega C_g), \tag{107}$$

where $(a)$ is from (106), and $(b)$ is from the fact that $\beta\tau_\beta C_\phi^2 \leq \frac{1}{4}$.

To prove (103) and (104), first note that

$$\|z_t - z_{t-\tau_\beta}\| \leq \sum_{j=t-\tau_\beta}^{t-1} \|z_{j+1} - z_j\|$$

$$\overset{(a)}{\leq} \sum_{j=t-\tau_\beta}^{t-1} \beta C_\phi^2\|z_j\| + \beta\tau_\beta(b_{\max} + L_\omega C_g)$$

$$\overset{(b)}{\leq} \sum_{j=t-\tau_\beta}^{t-1} \beta C_\phi^2(2\|z_{t-\tau_\beta}\| + 2\beta\tau_\beta(b_{\max} + L_\omega C_g)) + \beta\tau_\beta(b_{\max} + L_\omega C_g)$$

$$\leq \beta\tau_\beta C_\phi^2(2\|z_{t-\tau_\beta}\| + 2\beta\tau_\beta(b_{\max} + L_\omega C_g)) + \beta\tau_\beta(b_{\max} + L_\omega C_g)$$

$$= 2\beta\tau_\beta C_\phi^2\|z_{t-\tau_\beta}\| + (2\beta^2\tau_\beta^2 C_\phi^2 + \beta\tau_\beta)(b_{\max} + L_\omega C_g)$$

$$\overset{(c)}{\leq} 2\beta\tau_\beta C_\phi^2\|z_{t-\tau_\beta}\| + 2\beta\tau_\beta(b_{\max} + L_\omega C_g), \tag{108}$$

where $(a)$ is from (98), $(b)$ is from (107) and $(c)$ is due to the fact that $\beta\tau_\beta C_\phi^2 \leq \frac{1}{4}$. Moreover, it can be further shown that

$$\|z_t - z_{t-\tau_\beta}\| \leq 2\beta\tau_\beta C_\phi^2(\|z_t\| + \|z_t - z_{t-\tau_\beta}\|) + 2\beta\tau_\beta(b_{\max} + L_\omega C_g)$$

$$\leq 2\beta\tau_\beta C_\phi^2\|z_t\| + \frac{1}{2}\|z_t - z_{t-\tau_\beta}\| + 2\beta\tau_\beta(b_{\max} + L_\omega C_g), \tag{109}$$

where the last step is because $\beta\tau_\beta C_\phi^2 \leq \frac{1}{4}$. Hence

$$\|z_t - z_{t-\tau_\beta}\| \leq 4\beta\tau_\beta C_\phi^2\|z_t\| + 4\beta\tau_\beta(b_{\max} + L_\omega C_g). \tag{110}$$

$\square$

The bound on term $(b)$ in (91) is straightforward from the following lemma.

**Lemma 5.** *For any $t \geq \tau_\beta$, it follows that*

$$\left| \mathbb{E}\left[ z_t^\top \left( -A_{\theta_t} z_t - \frac{1}{\beta}(z_{t+1} - z_t) \right) \right] \right|$$
$$\leq (R_1 + R_3 + P_1 + P_2 + P_3)\mathbb{E}\left[ \|z_t\|^2 \right] + (Q_1 + Q_2 + Q_3 + P_1 + P_2 + P_3)$$
$$+ \frac{\alpha}{8\beta} L_\omega \mathbb{E}\left[ \|\nabla J(\theta_t)\|^2 \right], \tag{111}$$

*where the definition of $P_i, Q_i$ and $R_i$, $i = 1, 2, 3$, can be found in (114), (117) and (120).*

*Proof.* We only prove the case $t = \tau_\beta$ here. The proof for the general case with $t > \tau_\beta$ is similar, and thus is omitted here. First note that

$$\mathbb{E}\left[ z_{\tau_\beta}^\top \left( -A_{\theta_{\tau_\beta}} z_{\tau_\beta} - \frac{1}{\beta}\left( z_{\tau_\beta+1} - z_{\tau_\beta} \right) \right) \right]$$
$$= \mathbb{E}\left[ z_{\tau_\beta}^\top \left( -A_{\theta_{\tau_\beta}} + A_{\theta_{\tau_\beta}}\left( s_{\tau_\beta} \right) \right) z_{\tau_\beta} \right] - \mathbb{E}\left[ z_{\tau_\beta}^\top b_{\tau_\beta} \right] - \mathbb{E}\left[ z_{\tau_\beta}^\top \frac{\omega\left( \theta_{\tau_\beta} \right) - \omega\left( \theta_{\tau_\beta+1} \right)}{\beta} \right]. \tag{112}$$

We then bound the terms in (112) one by one. First, it can be shown that

$$\left| \mathbb{E}\left[ z_{\tau_\beta}^\top \left( -A_{\theta_{\tau_\beta}} + A_{\theta_{\tau_\beta}}\left( s_{\tau_\beta} \right) \right) z_{\tau_\beta} \right] \right|$$
$$\leq \left| \mathbb{E}\left[ z_0^\top \left( -A_{\theta_{\tau_\beta}} + A_{\theta_{\tau_\beta}}\left( s_{\tau_\beta} \right) \right) z_0 \right] \right| + \left| \mathbb{E}\left[ \left( z_{\tau_\beta} - z_0 \right)^\top \left( -A_{\theta_{\tau_\beta}} + A_{\theta_{\tau_\beta}}\left( s_{\tau_\beta} \right) \right) \left( z_{\tau_\beta} - z_0 \right) \right] \right|$$
$$+ 2\left| \mathbb{E}\left[ \left( z_{\tau_\beta} - z_0 \right)^\top \left( -A_{\theta_{\tau_\beta}} + A_{\theta_{\tau_\beta}}\left( s_{\tau_\beta} \right) \right) z_0 \right] \right|$$
$$\leq \|z_0\|^2 \left\| \mathbb{E}\left[ -A_{\theta_{\tau_\beta}} + A_{\theta_{\tau_\beta}}\left( s_{\tau_\beta} \right) \right] \right\| + 2C_\phi^2 \mathbb{E}\left[ \|z_{\tau_\beta} - z_0\|^2 \right] + 4\|z_0\|C_\phi^2 \mathbb{E}\left[ \|z_{\tau_\beta} - z_0\| \right]$$
$$\leq \|z_0\|^2 \left\| \mathbb{E}\left[ -A_{\theta_0} + A_{\theta_0}\left( s_{\tau_\beta} \right) \right] \right\| + \|z_0\|^2 \left\| \mathbb{E}\left[ -A_{\theta_0} + A_{\theta_{\tau_\beta}} \right] \right\|$$
$$+ \|z_0\|^2 \left\| \mathbb{E}\left[ -A_{\theta_{\tau_\beta}}\left( s_{\tau_\beta} \right) + A_{\theta_0}\left( s_{\tau_\beta} \right) \right] \right\| + 2C_\phi^2 \mathbb{E}\left[ \|z_{\tau_\beta} - z_0\|^2 \right] + 4\|z_0\|C_\phi^2 \mathbb{E}\left[ \|z_{\tau_\beta} - z_0\| \right]$$
$$\overset{(a)}{\leq} \left( \beta C_\phi^2 + 4C_\phi D_v C_g \alpha \tau_\beta \right) \|z_0\|^2 + 2C_\phi^2 \mathbb{E}\left[ \|z_{\tau_\beta} - z_0\|^2 \right] + 4\|z_0\|C_\phi^2 \mathbb{E}\left[ \|z_{\tau_\beta} - z_0\| \right], \tag{113}$$

where $(a)$ is due to the facts that $\left\| \mathbb{E}\left[ -A_{\theta_0} + A_{\theta_0}(s_{\tau_\beta}) \right] \right\| \leq C_\phi^2 \beta$ from the uniform ergodicity of the MDP, both $A_\theta$ and $A_\theta(s_{\tau_\beta})$ are Lipschitz with constant $2C_\phi D_v$, and $\|\theta_0 - \theta_{\tau_\beta}\| \leq \sum_{j=0}^{\tau_\beta-1} \|\theta_{j+1} - \theta_j\| \leq \alpha \tau_\beta C_g$.

We then plug in the results from Lemma 4, and hence we have that

$$\left| \mathbb{E}\left[ z_{\tau_\beta}^\top \left( -A_{\theta_{\tau_\beta}} + A_{\theta_{\tau_\beta}}\left( s_{\tau_\beta} \right) \right) z_{\tau_\beta} \right] \right|$$
$$\leq \left( \beta C_\phi^2 + 4C_\phi D_v C_g \alpha \tau_\beta \right) \|z_0\|^2 + 2C_\phi^2 \mathbb{E}\left[ \|z_{\tau_\beta} - z_0\|^2 \right] + 4\|z_0\|C_\phi^2 \mathbb{E}\left[ \|z_{\tau_\beta} - z_0\| \right]$$
$$\overset{(a)}{\leq} \left( \beta C_\phi^2 + 4C_\phi D_v C_g \alpha \tau_\beta \right) \left( 2(1 + 4\beta\tau_\beta C_\phi^2)^2 \mathbb{E}\left[ \|z_{\tau_\beta}\|^2 \right] + 32\beta^2 \tau_\beta^2 (b_{\max} + L_\omega C_g)^2 \right)$$
$$+ 2C_\phi^2 \left( 32\beta^2 \tau_\beta^2 C_\phi^4 \mathbb{E}\left[ \|z_{\tau_\beta}\|^2 \right] + 32\beta^2 \tau_\beta^2 (b_{\max} + L_\omega C_g)^2 \right)$$
$$+ 4C_\phi^2 \left( 4\beta\tau_\beta C_\phi^2 (1 + 4\beta\tau_\beta C_\phi^2)\mathbb{E}\left[ \|z_{\tau_\beta}\|^2 \right] + 4\beta\tau_\beta (b_{\max} + L_\omega C_g)(1 + 8\beta\tau_\beta C_\phi^2)\mathbb{E}\left[ \|z_{\tau_\beta}\| \right] \right)$$
$$+ 64C_\phi^2 \beta^2 \tau_\beta^2 (b_{\max} + L_\omega C_g)^2$$
$$\triangleq R_1 \mathbb{E}\left[ \|z_{\tau_\beta}\|^2 \right] + P_1 \mathbb{E}\left[ \|z_{\tau_\beta}\| \right] + Q_1, \tag{114}$$

where $(a)$ is from (104) and the fact that

$$\|z_0\| \leq \left\| z_{\tau_\beta} - z_0 \right\| + \left\| z_{\tau_\beta} \right\| \leq (1 + 4\beta\tau_\beta C_\phi^2) \left\| z_{\tau_\beta} \right\| + 4\beta\tau_\beta (b_{\max} + L_\omega C_g); \tag{115}$$

and $R_1 = 2(1 + 4\beta\tau_\beta C_\phi^2)^2 \left(\beta C_\phi^2 + 4C_\phi D_v C_g \alpha\tau_\beta\right) + 64\beta^2\tau_\beta^2 C_\phi^6 + 16\beta\tau_\beta C_\phi^4(1 + 4\beta\tau_\beta C_\phi^2) = \mathcal{O}(\beta\tau_\beta)$, $P_1 = 16C_\phi^2\beta\tau_\beta(b_{\max} + L_\omega C_g)(1 + 8\beta\tau_\beta C_\phi^2) = \mathcal{O}(\beta\tau_\beta)$ and $Q_1 = \left(\beta C_\phi^2 + 4C_\phi D_v C_g \alpha\tau_\beta\right)32\beta^2\tau_\beta^2(b_{\max}+L_\omega C_g)^2+64C_\phi^2\beta^2\tau_\beta^2(b_{\max}+L_\omega C_g)^2+64C_\phi^2\beta^2\tau_\beta^2(b_{\max}+L_\omega C_g)^2 = \mathcal{O}(\beta^2\tau^2)$.

Similarly, the second term in (112) can be bounded as follows

$$
\begin{aligned}
\left|\mathbb{E}\left[z_{\tau_\beta}^\top b_{\tau_\beta}(\theta_{\tau_\beta})\right]\right| &\leq \left|\mathbb{E}\left[(z_{\tau_\beta} - z_0)^\top b_{\tau_\beta}(\theta_{\tau_\beta})\right]\right| + \left|\mathbb{E}\left[z_0^\top b_{\tau_\beta}(\theta_0)\right]\right| \\
&\quad + \|\mathbb{E}\left[z_0^\top (b_{\tau_\beta}(\theta_{\tau_\beta}) - b_{\tau_\beta}(\theta_0))\right]\| \\
&\leq b_{\max}\mathbb{E}\left[\|z_{\tau_\beta} - z_0\|\right] + \beta b_{\max}\|z_0\| + \alpha\tau_\beta C_g L_b\|z_0\|,
\end{aligned}
\tag{116}
$$

where $L_b = 2C_\phi D_v R_\omega + L_\omega C_\phi^2 + \rho_{\max}((1 + \gamma)C_\phi^2 + D_v(r_{\max} + (1 + \gamma)C_v))$ is the Lipschitz constant of $b_t(\theta)$. Again applying Lemma 4 implies that

$$
\begin{aligned}
&\left|\mathbb{E}\left[z_{\tau_\beta}^\top b_{\tau_\beta}(\theta_{\tau_\beta})\right]\right| \\
&\leq b_{\max}\mathbb{E}\left[\|z_{\tau_\beta} - z_0\|\right] + \beta b_{\max}\|z_0\| + \alpha\tau_\beta C_g L_b\|z_0\| \\
&\leq b_{\max}\left(4\beta\tau_\beta C_\phi^2\mathbb{E}\left[\|z_{\tau_\beta}\|\right] + 4\beta\tau_\beta(b_{\max} + L_\omega C_g)\right) \\
&\quad + (\beta b_{\max} + \alpha\tau_\beta C_g L_b)\left(\left(1 + 4\beta\tau_\beta C_\phi^2\right)\mathbb{E}\left[\|z_{\tau_\beta}\|\right] + 4\beta\tau_\beta(b_{\max} + L_\omega C_g)\right) \\
&\triangleq P_2\mathbb{E}\left[\|z_{\tau_\beta}\|\right] + Q_2,
\end{aligned}
\tag{117}
$$

where $P_2 = 4\beta\tau_\beta b_{\max}C_\phi^2 + (\beta b_{\max}+\alpha\tau_\beta C_g L_b)\left(1 + 4\beta\tau_\beta C_\phi^2\right) = \mathcal{O}(\beta\tau_\beta)$ and $Q_2 = 4\beta\tau_\beta(b_{\max} + L_\omega C_g)(b_{\max} + \beta b_{\max} + \alpha\tau_\beta C_g L_b) = \mathcal{O}(\beta\tau_\beta)$.

We then bound the last term in (112) as follows

$$
\begin{aligned}
&\left|\mathbb{E}\left[z_{\tau_\beta}^\top \frac{\omega(\theta_{\tau_\beta}) - \omega(\theta_{\tau_\beta+1})}{\beta}\right]\right| \\
&\overset{(a)}{=} \left|\frac{1}{\beta}\mathbb{E}[z_{\tau_\beta}^\top \nabla\omega(\hat{\theta}_{\tau_\beta})(\theta_{\tau_\beta+1} - \theta_{\tau_\beta})]\right| \\
&= \left|\frac{\alpha}{\beta}\mathbb{E}[z_{\tau_\beta}^\top \nabla\omega(\hat{\theta}_{\tau_\beta})G_{\tau_\beta+1}(\theta_{\tau_\beta}, \omega_{\tau_\beta})]\right| \\
&= \left|\frac{\alpha}{\beta}\mathbb{E}\left[z_{\tau_\beta}^\top \nabla\omega(\hat{\theta}_{\tau_\beta})\left(G_{\tau_\beta+1}(\theta_{\tau_\beta}, \omega_{\tau_\beta}) - G_{\tau_\beta+1}(\theta_{\tau_\beta}, \omega(\theta_{\tau_\beta})) + G_{\tau_\beta+1}(\theta_{\tau_\beta}, \omega(\theta_{\tau_\beta}))\right.\right.\right. \\
&\quad \left.\left.\left. + \frac{\nabla J(\theta_{\tau_\beta})}{2} - \frac{\nabla J(\theta_{\tau_\beta})}{2}\right)\right]\right| \\
&= \left|\frac{\alpha}{\beta}\mathbb{E}\left[z_{\tau_\beta}^\top \nabla\omega(\hat{\theta}_{\tau_\beta})(G_{\tau_\beta+1}(\theta_{\tau_\beta}, \omega_{\tau_\beta}) - G_{\tau_\beta+1}(\theta_{\tau_\beta}, \omega(\theta_{\tau_\beta})))\right]\right| \\
&\quad + \left|\frac{\alpha}{\beta}\mathbb{E}\left[z_{\tau_\beta}^\top \nabla\omega(\hat{\theta}_{\tau_\beta})\left(G_{\tau_\beta+1}(\theta_{\tau_\beta}, \omega(\theta_{\tau_\beta})) + \frac{\nabla J(\theta_{\tau_\beta})}{2}\right)\right]\right| \\
&\quad + \left|\frac{\alpha}{\beta}\mathbb{E}\left[z_{\tau_\beta}^\top \nabla\omega(\hat{\theta}_{\tau_\beta})\left(-\frac{\nabla J(\theta_{\tau_\beta})}{2}\right)\right]\right| \\
&\overset{(b)}{\leq} \frac{\alpha}{\beta}L_\omega L_g\mathbb{E}\left[\|z_{\tau_\beta}\|^2\right] + \frac{\alpha}{2\beta}L_\omega\mathbb{E}\left[\|z_{\tau_\beta}\|^2\right] + \frac{\alpha}{8\beta}L_\omega\mathbb{E}\left[\|\nabla J(\theta_{\tau_\beta})\|^2\right] \\
&\quad + \frac{\alpha}{\beta}\left|\mathbb{E}\left[z_{\tau_\beta}^\top \nabla\omega(\theta_{\tau_\beta})\left(G_{\tau_\beta+1}(\theta_{\tau_\beta}, \omega(\theta_{\tau_\beta})) + \frac{\nabla J(\theta_{\tau_\beta})}{2}\right)\right]\right| \\
&\quad + \frac{\alpha}{\beta}\left|\mathbb{E}\left[z_{\tau_\beta}^\top (\nabla\omega(\hat{\theta}_{\tau_\beta}) - \nabla\omega(\theta_{\tau_\beta}))\left(G_{\tau_\beta+1}(\theta_{\tau_\beta}, \omega(\theta_{\tau_\beta})) + \frac{\nabla J(\theta_{\tau_\beta})}{2}\right)\right]\right|
\end{aligned}
$$

$$\leq \frac{\alpha}{\beta} L_\omega L_g \mathbb{E}\left[\left\|z_{\tau_\beta}\right\|^2\right] + \frac{\alpha}{2\beta} L_\omega \mathbb{E}\left[\left\|z_{\tau_\beta}\right\|^2\right] + \frac{\alpha}{8\beta} L_\omega \mathbb{E}\left[\left\|\nabla J(\theta_{\tau_\beta})\right\|^2\right]$$

$$+ \frac{\alpha}{\beta}\left|\mathbb{E}\left[z_0^\top \nabla\omega(\theta_{\tau_\beta})\left(G_{\tau_\beta+1}(\theta_{\tau_\beta},\omega(\theta_{\tau_\beta})) + \frac{\nabla J(\theta_{\tau_\beta})}{2}\right)\right]\right|$$

$$+ \frac{\alpha}{\beta}\left|\mathbb{E}\left[(z_{\tau_\beta}-z_0)^\top \nabla\omega(\theta_{\tau_\beta})\left(G_{\tau_\beta+1}(\theta_{\tau_\beta},\omega(\theta_{\tau_\beta})) + \frac{\nabla J(\theta_{\tau_\beta})}{2}\right)\right]\right|$$

$$+ \frac{2\alpha}{\beta} C_g D_\omega \mathbb{E}\left[\left\|z_{\tau_\beta}\right\| \left\|\theta_{\tau_\beta} - \theta_{\tau_\beta+1}\right\|\right]$$

$$\leq \frac{\alpha}{\beta} L_\omega L_g \mathbb{E}\left[\left\|z_{\tau_\beta}\right\|^2\right] + \frac{\alpha}{2\beta} L_\omega \mathbb{E}\left[\left\|z_{\tau_\beta}\right\|^2\right] + \frac{\alpha}{8\beta} L_\omega \mathbb{E}\left[\left\|\nabla J(\theta_{\tau_\beta})\right\|^2\right]$$

$$+ \frac{\alpha}{\beta}\left|\mathbb{E}\left[z_0^\top \nabla\omega(\theta_0)\left(G_{\tau_\beta+1}(\theta_0,\omega(\theta_0)) + \frac{\nabla J(\theta_0)}{2}\right)\right]\right|$$

$$+ \frac{\alpha}{\beta}\left|\mathbb{E}\left[z_0^\top \left(\nabla\omega(\theta_{\tau_\beta})\left(G_{\tau_\beta+1}(\theta_{\tau_\beta},\omega(\theta_{\tau_\beta})) + \frac{\nabla J(\theta_{\tau_\beta})}{2}\right)\right.\right.\right.$$

$$\left.\left.\left. - \nabla\omega(\theta_0)\left(G_{\tau_\beta+1}(\theta_0,\omega(\theta_0)) + \frac{\nabla J(\theta_0)}{2}\right)\right)\right]\right|$$

$$+ \frac{\alpha}{\beta}\left|\mathbb{E}\left[(z_{\tau_\beta}-z_0)^\top \nabla\omega(\theta_{\tau_\beta})\left(G_{\tau_\beta+1}(\theta_{\tau_\beta},\omega(\theta_{\tau_\beta}) + \frac{\nabla J(\theta_{\tau_\beta})}{2}\right)\right]\right|$$

$$+ \frac{2\alpha}{\beta} C_g D_\omega \mathbb{E}\left[\left\|z_{\tau_\beta}\right\| \left\|\theta_{\tau_\beta} - \theta_{\tau_\beta+1}\right\|\right]$$

$$\leq \frac{\alpha}{\beta} L_\omega L_g \mathbb{E}\left[\left\|z_{\tau_\beta}\right\|^2\right] + \frac{\alpha}{8\beta} L_\omega \mathbb{E}\left[\left\|\nabla J(\theta_{\tau_\beta})\right\|^2\right]$$

$$+ \frac{\alpha}{\beta}\|z_0\|L_\omega \left\|\mathbb{E}\left[G_{\tau_\beta+1}(\theta_0,\omega(\theta_0)) + \frac{\nabla J(\theta_{\tau_\beta})}{2}\right]\right\|$$

$$+ \frac{\alpha}{2\beta} L_\omega \mathbb{E}\left[\left\|z_{\tau_\beta}\right\|^2\right] + \frac{\alpha}{\beta}\|z_0\|L_k \mathbb{E}\left[\left\|\theta_{\tau_\beta} - \theta_0\right\|\right] + \frac{2\alpha}{\beta} L_\omega C_g \mathbb{E}\left[\left\|z_{\tau_\beta} - z_0\right\|\right]$$

$$+ \frac{2\alpha}{\beta} C_g D_\omega \mathbb{E}\left[\left\|z_{\tau_\beta}\right\| \left\|\theta_{\tau_\beta} - \theta_{\tau_\beta+1}\right\|\right]$$

$$\overset{(c)}{\leq} \frac{\alpha}{\beta} L_\omega L_g \mathbb{E}\left[\left\|z_{\tau_\beta}\right\|^2\right] + \frac{\alpha}{2\beta} L_\omega \mathbb{E}\left[\left\|z_{\tau_\beta}\right\|^2\right] + \frac{\alpha}{8\beta} L_\omega \mathbb{E}\left[\left\|\nabla J(\theta_{\tau_\beta})\right\|^2\right] + \frac{\alpha}{\beta}\|z_0\|L_\omega C_g \beta$$

$$+ \frac{\alpha^2}{\beta}\tau_\beta L_k \|z_0\| C_g + \frac{2\alpha}{\beta} L_\omega C_g \mathbb{E}\left[\left\|z_{\tau_\beta} - z_0\right\|\right] + \frac{2\alpha^2}{\beta} C_g^2 D_\omega \mathbb{E}\left[\left\|z_{\tau_\beta}\right\|\right]$$

$$= \left(\frac{\alpha}{\beta} L_\omega L_g + \frac{\alpha}{2\beta} L_\omega\right)\mathbb{E}\left[\left\|z_{\tau_\beta}\right\|^2\right] + \frac{2\alpha^2}{\beta} C_g^2 D_\omega \mathbb{E}\left[\left\|z_{\tau_\beta}\right\|\right] + \frac{\alpha}{8\beta} L_\omega \mathbb{E}\left[\left\|\nabla J(\theta_{\tau_\beta})\right\|^2\right]$$

$$+ \left(\alpha L_\omega C_g + \frac{\alpha^2}{\beta}\tau_\beta L_k C_g\right)\|z_0\| + \frac{2\alpha}{\beta} L_\omega C_g \mathbb{E}\left[\left\|z_{\tau_\beta} - z_0\right\|\right], \tag{118}$$

where $(a)$ is from the Mean-Value theorem and $\hat{\theta}_{\tau_\beta} = c\theta_{\tau_\beta} + (1-c)\theta_{\tau_\beta+1}$ for some $c \in [0,1]$, $(b)$ is from Lemmas 1 and 2, $(c)$ is due to the fact that $\left\|\mathbb{E}\left[G_{t+1}(\theta_0,\omega(\theta_0)) + \frac{\nabla J(\theta_0)}{2}\right]\right\| \leq C_g \beta$ for any $t \geq \tau_\beta$ and $\|\theta_{\tau_\beta} - \theta_0\| \leq \alpha\tau_\beta C_g$, and $L_k = 2C_g D_\omega + \left(L_J + \frac{L_g'}{2}\right)L_\omega$ is the Lipschitz constant of $\nabla\omega(\theta)\left(G_{t+1}(\theta,\omega(\theta)) + \frac{\nabla J(\theta)}{2}\right)$, and $L_g'$ is the Lipschitz constant of $G_{t+1}(\theta,\omega(\theta))$.

Our next step is to rewrite the bound in (118) using $\|z_{\tau_\beta}\|$. Note that from Lemma 4, we have that

$$\left\|z_0\right\| \leq \left\|z_{\tau_\beta} - z_0\right\| + \left\|z_{\tau_\beta}\right\| \leq \left(1 + 4\beta\tau_\beta C_\phi^2\right)\left\|z_{\tau_\beta}\right\| + 4\beta\tau_\beta(b_{\max} + L_\omega C_g). \tag{119}$$

Plugging in (118), it follows that

$$\left|\mathbb{E}\left[z_{\tau_\beta}^\top \frac{\omega(\theta_{\tau_\beta}) - \omega(\theta_{\tau_\beta+1})}{\beta}\right]\right|$$

$$\leq \left(\frac{\alpha}{\beta}L_\omega L_g + \frac{\alpha}{2\beta}L_\omega\right)\mathbb{E}\left[\left\|z_{\tau_\beta}\right\|^2\right] + \frac{2\alpha^2}{\beta}C_g^2 D_\omega \mathbb{E}\left[\left\|z_{\tau_\beta}\right\|\right] + \frac{\alpha}{8\beta}L_\omega\mathbb{E}\left[\left\|\nabla J(\theta_{\tau_\beta})\right\|^2\right]$$

$$+ \left(\alpha L_\omega C_g + \frac{\alpha^2}{\beta}\tau_\beta L_k C_g\right)\|z_0\| + \frac{2\alpha}{\beta}L_\omega C_g\mathbb{E}\left[\left\|z_{\tau_\beta} - z_0\right\|\right]$$

$$\leq \left(\frac{\alpha}{\beta}L_\omega L_g + \frac{\alpha}{2\beta}L_\omega\right)\mathbb{E}\left[\left\|z_{\tau_\beta}\right\|^2\right] + \frac{2\alpha^2}{\beta}C_g^2 D_\omega \mathbb{E}\left[\left\|z_{\tau_\beta}\right\|\right] + \frac{\alpha}{8\beta}L_\omega\mathbb{E}\left[\left\|\nabla J(\theta_{\tau_\beta})\right\|^2\right]$$

$$+ \left(\alpha L_\omega C_g + \frac{\alpha^2}{\beta}\tau_\beta L_k C_g\right)\left((1 + 4\beta\tau_\beta C_\phi^2)\mathbb{E}\left[\left\|z_{\tau_\beta}\right\|\right] + 4\beta\tau_\beta(b_{\max} + L_\omega C_g)\right)$$

$$+ \frac{2\alpha}{\beta}L_\omega C_g\left(\mathbb{E}\left[4\beta\tau_\beta C_\phi^2\left\|z_{\tau_\beta}\right\|\right] + 4\beta\tau_\beta(b_{\max} + L_\omega C_g)\right)$$

$$= \left(\frac{\alpha}{\beta}L_\omega L_g + \frac{\alpha}{2\beta}L_\omega\right)\mathbb{E}\left[\left\|z_{\tau_\beta}\right\|^2\right]$$

$$+ \left(\frac{2\alpha^2}{\beta}C_g^2 D_\omega + \left(\alpha L_\omega C_g + \frac{\alpha^2}{\beta}\tau_\beta L_k C_g\right)(1 + 4\beta\tau_\beta C_\phi^2) + 8\alpha\tau_\beta L_\omega C_g C_\phi^2\right)\mathbb{E}\left[\left\|z_{\tau_\beta}\right\|\right]$$

$$+ \frac{\alpha}{8\beta}L_\omega\mathbb{E}\left[\left\|\nabla J(\theta_{\tau_\beta})\right\|^2\right] + \left(\alpha L_\omega C_g + \frac{\alpha^2}{\beta}\tau_\beta L_k C_g\right)(4\beta\tau_\beta(b_{\max} + L_\omega C_g))$$

$$+ 8\alpha\tau_\beta L_\omega C_g(b_{\max} + L_\omega C_g)$$

$$\triangleq R_3\mathbb{E}\left[\left\|z_{\tau_\beta}\right\|^2\right] + P_3\mathbb{E}\left[\left\|z_{\tau_\beta}\right\|\right] + Q_3 + \frac{\alpha}{8\beta}L_\omega\mathbb{E}\left[\left\|\nabla J(\theta_{\tau_\beta})\right\|^2\right], \tag{120}$$

where $R_3 = \left(\frac{\alpha}{\beta}L_\omega L_g + \frac{\alpha}{2\beta}L_\omega\right) = \mathcal{O}\left(\frac{\alpha}{\beta}\right)$, $P_3 = \left(\frac{2\alpha^2}{\beta}C_g^2 D_\omega + \left(\alpha L_\omega C_g + \frac{\alpha^2}{\beta}\tau_\beta L_k C_g\right)(1 + 4\beta\tau_\beta C_\phi^2) + 8\alpha\tau_\beta L_\omega C_g C_\phi^2\right) = \mathcal{O}(\alpha\tau_\beta)$ and $Q_3 = \left(\alpha L_\omega C_g + \frac{\alpha^2}{\beta}\tau_\beta L_k C_g\right)(4\beta\tau_\beta(b_{\max} + L_\omega C_g)) + 8\alpha\tau_\beta L_\omega C_g(b_{\max} + L_\omega C_g) = \mathcal{O}(\alpha\tau_\beta)$.

Then we combine all three bounds in (114), (117) and (120), and it follows that

$$\left|\mathbb{E}\left[z_{\tau_\beta}^\top\left(-A_{\theta_{\tau_\beta}}z_{\tau_\beta} - \frac{1}{\beta}\left(z_{\tau_\beta+1} - z_{\tau_\beta}\right)\right)\right]\right|$$

$$\leq (R_1 + R_3)\mathbb{E}\left[\left\|z_{\tau_\beta}\right\|^2\right] + (P_1 + P_2 + P_3)\mathbb{E}\left[\left\|z_{\tau_\beta}\right\|\right] + (Q_1 + Q_2 + Q_3)$$

$$+ \frac{\alpha}{8\beta}L_\omega\mathbb{E}\left[\left\|\nabla J(\theta_{\tau_\beta})\right\|^2\right], \tag{121}$$

Finally due to the fact that $x \leq x^2 + 1, \forall x \in \mathbb{R}$, it follows that

$$\left|\mathbb{E}\left[z_{\tau_\beta}^\top\left(-A_{\theta_{\tau_\beta}}z_{\tau_\beta} - \frac{1}{\beta}\left(z_{\tau_\beta+1} - z_{\tau_\beta}\right)\right)\right]\right|$$

$$\leq (R_1 + R_3 + P_1 + P_2 + P_3)\mathbb{E}\left[\left\|z_{\tau_\beta}\right\|^2\right] + (Q_1 + Q_2 + Q_3 + P_1 + P_2 + P_3)$$

$$+ \frac{\alpha}{8\beta}L_\omega\mathbb{E}\left[\left\|\nabla J(\theta_{\tau_\beta})\right\|^2\right]. \tag{122}$$

This completes the proof. $\qquad\square$

## C.2 Proof under the Markovian Setting

In this section, we prove Theorem 1 under the Markovian setting.

From the $L_J$-smoothness of $J(\theta)$, it follows that

$$J(\theta_{t+1}) \leq J(\theta_t) + \langle\nabla J(\theta_t), \theta_{t+1} - \theta_t\rangle + \frac{L_J}{2}\|\theta_{t+1} - \theta_t\|^2$$

$$= J(\theta_t) + \alpha\langle\nabla J(\theta_t), G_{t+1}(\theta_t, \omega_t)\rangle + \frac{L_J}{2}\alpha^2\|G_{t+1}(\theta_t, \omega_t)\|^2$$

$$= J(\theta_t) - \alpha \left\langle \nabla J(\theta_t), -G_{t+1}(\theta_t, \omega_t) - \frac{\nabla J(\theta_t)}{2} + G_{t+1}(\theta_t, \omega(\theta_t)) - G_{t+1}(\theta_t, \omega(\theta_t)) \right\rangle$$

$$- \frac{\alpha}{2} \|\nabla J(\theta_t)\|^2 + \frac{L_J}{2} \alpha^2 \|G_{t+1}(\theta_t, \omega_t)\|^2$$

$$= J(\theta_t) - \alpha \left\langle \nabla J(\theta_t), -G_{t+1}(\theta_t, \omega_t) + G_{t+1}(\theta_t, \omega(\theta_t)) \right\rangle$$

$$+ \alpha \left\langle \nabla J(\theta_t), \frac{\nabla J(\theta_t)}{2} + G_{t+1}(\theta_t, \omega(\theta_t)) \right\rangle - \frac{\alpha}{2} \|\nabla J(\theta_t)\|^2 + \frac{L_J}{2} \alpha^2 \|G_{t+1}(\theta_t, \omega_t)\|^2$$

$$\leq J(\theta_t) + \alpha L_g \|\nabla J(\theta_t)\| \|\omega(\theta_t) - \omega_t\| + \alpha \left\langle \nabla J(\theta_t), \frac{\nabla J(\theta_t)}{2} + G_{t+1}(\theta_t, \omega(\theta_t)) \right\rangle$$

$$- \frac{\alpha}{2} \|\nabla J(\theta_t)\|^2 + \frac{L_J}{2} \alpha^2 \|G_{t+1}(\theta_t, \omega_t)\|^2$$

$$\overset{(a)}{\leq} J(\theta_t) + \alpha L_g \|\nabla J(\theta_t)\| \|z_t\| + \alpha \left\langle \nabla J(\theta_t), \frac{\nabla J(\theta_t)}{2} + G_{t+1}(\theta_t, \omega(\theta_t)) \right\rangle$$

$$- \frac{\alpha}{2} \|\nabla J(\theta_t)\|^2 + \frac{L_J}{2} \alpha^2 C_g^2, \tag{123}$$

where $(a)$ is from the fact that $\|\theta_{t+1} - \theta_t\| \leq \alpha C_g$. Thus by re-arranging the terms, taking expectation and summing up w.r.t. $t$ from 0 to $T-1$, it follows that

$$\frac{\alpha}{2} \sum_{t=0}^{T-1} \mathbb{E}[\|\nabla J(\theta_t)\|^2]$$

$$\leq -\mathbb{E}[J(\theta_T)] + J(\theta_0) + \alpha L_g \sqrt{\sum_{t=0}^{T-1} \mathbb{E}[\|\nabla J(\theta_t)\|^2]} \sqrt{\sum_{t=0}^{T-1} \mathbb{E}[\|z_t\|^2]} + \sum_{t=0}^{T-1} \alpha \mathbb{E}[\zeta_G(\theta_t, O_t)]$$

$$+ L_J \alpha^2 T C_g^2, \tag{124}$$

where $\zeta_G(\theta_t, O_t) = \left\langle \nabla J(\theta_t), \frac{\nabla J(\theta_t)}{2} + G_{t+1}(\theta_t, \omega(\theta_t)) \right\rangle$. We then bound $\zeta_G$ in the following lemma.

**Lemma 6.** *For any $t \geq \tau_\beta$,*

$$\mathbb{E}[\zeta_G(\theta_t, O_t)] \leq 2C_g^2 \beta + 2\alpha \tau_\beta L_\zeta C_g. \tag{125}$$

*Proof.* We only need to consider the case $t = \tau_\beta$, the proof for general case of $t \geq \tau_\beta$ is similar, and thus is omitted here. We first have that

$$\zeta_G(\theta_{\tau_\beta}, O_{\tau_\beta}) = \left\langle \nabla J(\theta_{\tau_\beta}), \frac{\nabla J(\theta_{\tau_\beta})}{2} + G_{\tau_\beta+1}(\theta_{\tau_\beta}, \omega(\theta_{\tau_\beta})) \right\rangle$$

$$= \left\langle \nabla J(\theta_0), \frac{\nabla J(\theta_0)}{2} + G_{\tau_\beta+1}(\theta_0, \omega(\theta_0)) \right\rangle$$

$$+ \left\langle \nabla J(\theta_{\tau_\beta}), \frac{\nabla J(\theta_{\tau_\beta})}{2} + G_{\tau_\beta+1}(\theta_{\tau_\beta}, \omega(\theta_{\tau_\beta})) \right\rangle$$

$$- \left\langle \nabla J(\theta_0), \frac{\nabla J(\theta_0)}{2} + G_{\tau_\beta+1}(\theta_0, \omega(\theta_0)) \right\rangle$$

$$\leq \left\langle \nabla J(\theta_0), \frac{\nabla J(\theta_0)}{2} + G_{\tau_\beta+1}(\theta_0, \omega(\theta_0)) \right\rangle + 2L_\zeta \|\theta_{\tau_\beta} - \theta_0\|$$

$$\leq \left\langle \nabla J(\theta_0), \frac{\nabla J(\theta_0)}{2} + G_{\tau_\beta+1}(\theta_0, \omega(\theta_0)) \right\rangle + 2\alpha \tau_\beta L_\zeta C_g, \tag{126}$$

where $L_\zeta = 2C_g(L_g' + \frac{3L_J}{2})$ is the Lipschitz constant of $\zeta_G(\theta, O_t)$.

Then it follows that

$$\mathbb{E}[\zeta_G(\theta_{\tau_\beta}, O_{\tau_\beta})]$$

$$= \mathbb{E}\left[\left\langle \nabla J(\theta_0), \frac{\nabla J(\theta_0)}{2} + G_{\tau_\beta+1}(\theta_0, \omega(\theta_0))\right\rangle\right] + 2\alpha L_\zeta C_g \tau_\beta$$
$$\leq 2C_g^2 \beta + 2\alpha \tau_\beta L_\zeta C_g, \tag{127}$$

where the last step follows from the uniform ergodicity of the MDP (Assumption 4). $\qquad\square$

Plugging the bound in (124), it follows that

$$\frac{\alpha}{2} \sum_{t=0}^{T-1} \mathbb{E}[\|\nabla J(\theta_t)\|^2]$$

$$\leq J(\theta_0) - J^* + \alpha L_g \sqrt{\sum_{t=0}^{T-1} \mathbb{E}[\|\nabla J(\theta_t)\|^2]} \sqrt{\sum_{t=0}^{T-1} \mathbb{E}[\|z_t\|^2]}$$
$$+ \alpha^2 C_g^2 L_J T + \alpha \left(T(2C_g^2 \beta + 2\alpha \tau_\beta L_\zeta C_g) + 4\tau_\beta C_g^2\right), \tag{128}$$

and thus

$$\sum_{t=0}^{T-1} \mathbb{E}[\|\nabla J(\theta_t)\|^2]$$

$$\leq \frac{2(J(\theta_0) - J^*)}{\alpha} + 2L_g \sqrt{\sum_{t=0}^{T-1} \mathbb{E}[\|\nabla J(\theta_t)\|^2]} \sqrt{\sum_{t=0}^{T-1} \mathbb{E}[\|z_t\|^2]} + 2\alpha C_g^2 L_J T$$
$$+ 2\left(T(2C_g^2 \beta + 2\alpha \tau_\beta L_\zeta C_g) + 4\tau_\beta C_g^2\right). \tag{129}$$

This further implies that

$$\frac{\sum_{t=0}^{T-1} \mathbb{E}[\|\nabla J(\theta_t)\|^2]}{T}$$

$$\leq \frac{2(J(\theta_0) - J^*)}{\alpha T} + 2L_g \sqrt{\frac{\sum_{t=0}^{T-1} \mathbb{E}[\|\nabla J(\theta_t)\|^2]}{T}} \sqrt{\frac{\sum_{t=0}^{T-1} \mathbb{E}[\|z_t\|^2]}{T}} + 2\alpha C_g^2 L_J$$
$$+ 2\left((2C_g^2 \beta + 2\alpha \tau_\beta L_\zeta C_g) + 4C_g^2 \frac{\tau_\beta}{T}\right). \tag{130}$$

We plug in the tracking error (97), and it follows that

$$\frac{\sum_{t=0}^{T-1} \mathbb{E}[\|\nabla J(\theta_t)\|^2]}{T}$$

$$\leq \frac{2(J(\theta_0) - J^*)}{\alpha T} + 2\alpha C_g^2 L_J + 2\left((2C_g^2 \beta + 2\alpha \tau_\beta L_\zeta C_g) + 4C_g^2 \frac{\tau_\beta}{T}\right)$$

$$+ 2L_g \sqrt{\frac{\sum_{t=0}^{T-1} \mathbb{E}[\|\nabla J(\theta_t)\|^2]}{T}}$$

$$\cdot \sqrt{(2\|z_0\| + 2\beta\tau_\beta(b_{\max} + L_\omega C_g))^2 \left(\frac{1}{Tq} + \frac{\tau_\beta}{T}\right) + \frac{\alpha L_\omega}{4q} \frac{\sum_{t=0}^{T-1} \mathbb{E}[\|\nabla J(\theta_t)\|^2]}{T} + \frac{p}{q}}$$

$$\leq \frac{2(J(\theta_0) - J^*)}{\alpha T} + 2\alpha C_g^2 L_J + 2\left((2C_g^2 \beta + 2\alpha \tau_\beta L_\zeta C_g) + 4C_g^2 \frac{\tau_\beta}{T}\right)$$

$$+ 2L_g \sqrt{\frac{\alpha L_\omega}{4q} \frac{\sum_{t=0}^{T-1} \mathbb{E}[\|\nabla J(\theta_t)\|^2]}{T}}$$

$$+ 2L_g \sqrt{\frac{\sum_{t=0}^{T-1} \mathbb{E}[\|\nabla J(\theta_t)\|^2]}{T}} \sqrt{(2\|z_0\| + 2\beta\tau_\beta(b_{\max} + L_\omega C_g))^2 \left(\frac{1}{Tq} + \frac{\tau_\beta}{T}\right) + \frac{p}{q}}. \tag{131}$$

Note that $2L_g \sqrt{\frac{\alpha L_\omega}{4q}} = \mathcal{O}\left(\sqrt{\frac{\alpha}{\beta}}\right)$, hence we can choose $\alpha$ and $\beta$ such that $2L_g \sqrt{\frac{\alpha L_\omega}{4q}} \leq \frac{1}{2}$. Hence it follows that

$$\frac{\sum_{t=0}^{T-1} \mathbb{E}[\|\nabla J(\theta_t)\|^2]}{T} \leq \frac{4(J(\theta_0) - J^*)}{\alpha T} + 4\alpha C_g^2 L_J + 4\left((2C_g^2 \beta + 2\alpha \tau_\beta L_\zeta C_g) + 4C_g^2 \frac{\tau_\beta}{T}\right)$$

$$+ 4L_g \sqrt{\frac{\sum_{t=0}^{T-1} \mathbb{E}[\|\nabla J(\theta_t)\|^2]}{T}}$$

$$\cdot \sqrt{(2\|z_0\| + 2\beta\tau_\beta(b_{\max} + L_\omega C_g))^2 \left(\frac{1}{Tq} + \frac{\tau_\beta}{T}\right) + \frac{p}{q}}$$

$$\triangleq U \sqrt{\frac{\sum_{t=0}^{T-1} \mathbb{E}[\|\nabla J(\theta_t)\|^2]}{T}} + V, \tag{132}$$

where $U = 4L_g \sqrt{(2\|z_0\| + 2\beta\tau_\beta(b_{\max} + L_\omega C_g))^2 \left(\frac{1}{Tq} + \frac{\tau_\beta}{T}\right) + \frac{p}{q}} = \mathcal{O}\left(\sqrt{\beta\tau_\beta + \frac{1}{T\beta}}\right)$ and $V = \frac{4(J(\theta_0) - J^*)}{\alpha T} + 4\alpha C_g^2 L_J + 4\left((2C_g^2\beta + 2\alpha\tau_\beta L_\zeta C_g) + 4C_g^2\frac{\tau_\beta}{T}\right) = \mathcal{O}\left(\frac{1}{T\alpha} + \alpha\tau_\beta + \beta\right)$. Thus it can be shown that

$$\frac{\sum_{t=0}^{T-1} \mathbb{E}[\|\nabla J(\theta_t)\|^2]}{T} \leq \left(\frac{U + \sqrt{U^2 + 4V}}{2}\right)^2$$

$$\leq U^2 + 2V$$

$$= 16L_g^2 \left((2\|z_0\| + 2\beta\tau_\beta(b_{\max} + L_\omega C_g))^2 \left(\frac{1}{Tq} + \frac{\tau_\beta}{T}\right) + \frac{p}{q}\right)$$

$$+ \frac{8(J(\theta_0) - J^*)}{\alpha T} + 8\alpha C_g^2 L_J + 8\left((2C_g^2\beta + 2\alpha\tau_\beta L_\zeta C_g) + 4C_g^2\frac{\tau_\beta}{T}\right)$$

$$= \mathcal{O}\left(\beta\tau_\beta + \frac{1}{T\beta} + \alpha\tau_\beta + \frac{1}{T\alpha}\right). \tag{133}$$

This completes the proof.

### C.3 Choice of Step-sizes

In the proof under the Markovian setting, we first assume $\beta\tau_\beta C_\phi^2 \leq \frac{1}{4}$. The last assumption on the step-sizes is $\frac{\alpha}{q} \leq \frac{1}{4L_g^2 L_\omega}$, where $q = 2\beta\lambda_v - 2\beta(R_1 + R_3 + P_1 + P_2 + P_3) - 2\beta^2 C_\phi^4 = \mathcal{O}(\beta)$. Note that this assumption can be satisfied by controlling $\frac{\alpha}{\beta}$ similar to Section B.3, which we omit here. Hence we set $\beta < \min\left\{1, \frac{1}{4\tau_\beta C_\phi^2}\right\}$, and $\frac{\alpha}{q} \leq \left\{1, \frac{1}{4L_g^2 L_\omega}\right\}$.

## D Experiments

In this section, we provide some numerical experiments on two RL examples: the Garnet problem [Archibald et al., 1995] and the "spiral" counter example in [Tsitsiklis and Van Roy, 1997].

### D.1 Garnet Problem

The first experiment is on the Garnet problem [Archibald et al., 1995], which can be characterized by $\mathcal{G}(|\mathcal{S}|, |\mathcal{A}|, b, N)$. Here $b$ is a branching parameter specifying how many next states are possible for each state-action pair, and these $b$ states are chosen uniformly at random. The transition probabilities are generated by sampling uniformly and randomly between 0 and 1. The parameter $N$ is the dimension of $\theta$ to be updated. In our experiments, we generate a reward matrix uniformly and randomly between 0 and 1. For every state $s$ we randomly generate one feature function $k(s) \in [0, 1]$ using as the input. In both experiments, we use a five-layer neural network with (1,2,2,3,1) neurons in each layer as the function approximator. And for the activation function, we use the Sigmoid function, i.e., $f(x) = \frac{1}{1+e^{-x}}$. We set all the weights and bias of the neurons as the parameter $\theta \in \mathbb{R}^{23}$.

We consider two sets of parameters: $\mathcal{G}(5, 2, 5, 23)$ and $\mathcal{G}(3, 2, 3, 23)$. We set the step-size $\alpha = 0.01$ and $\beta = 0.05$, and also the discount factor $\gamma = 0.95$. In Figures 1 and 2, we plot the squared gradient norm v.s. the number of samples using 40 Garnet MDP trajectories, i.e., at each time $t$, we plot $\|\nabla J(\theta_t)\|^2$. The upper and lower envelopes of the curves correspond to the 95 and 5 percentiles of the 40 curves, respectively. We also plot the estimated variance of the stochastic update along the

iterations in Figures 1(b) and 2(b). Specifically, we first run the algorithm to get a sequence of $\theta_t$ and $\omega_t$. Then we generate 500 different trajectories $O^i = (O_1^i, O_2^i, ..., O_t^i, ...)$ where $i = 1, ..., 500$, and use them to estimate the variance $\|G_{t+1}^i(\theta_t, \omega_t) - \nabla J(\theta_t)\|^2$ and plot $\frac{\sum_{i=1}^{500} \|G_{t+1}^i(\theta_t, \omega_t) - \nabla J(\theta_t)\|^2}{500}$ at each time $t$.

It can be seen from the figures that both gradient norm $\|\nabla J(\theta_t)\|$ and the estimated variance converge to zero.

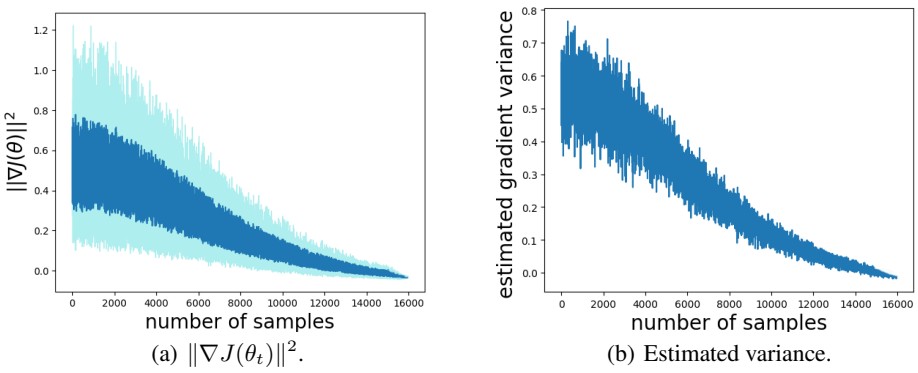

(a) $\|\nabla J(\theta_t)\|^2$.        (b) Estimated variance.

Figure 1: Garnet problem 1: $\mathcal{G}(5, 2, 5, 23)$.

## D.2 Spiral Counter Example

In our second experiment, we consider the spiral counter example proposed in [Tsitsiklis and Van Roy, 1997], which is often used to show the TD algorithm may diverge with nonlinear function approximation. The problem setting is given in Figure 3. There are three states and each state can transit to the next one with probability $\frac{1}{2}$ or stay at the current state with probability $\frac{1}{2}$. The reward is always zero with the discount factor $\gamma = 0.9$. Similar to [Bhatnagar et al., 2009], we consider the value function approximation:

$$V_\theta(s) = (a(s)\cos(k\theta) + b(s)\sin(k\theta))e^{\epsilon\theta}, \tag{134}$$

where in Figure 4, $a = [0.94, -0.43, 0.18]$ and $b = [0.21, -0.52, 0.76]$; and in Figure 5, $a = [0.21, -0.33, 0.29]$ and $b = [0.68, 0.41, 0.82]$. We let $k = 0.866$ and $\epsilon = 0.1$. The step-size are chosen as $\alpha = 0.01$ and $\beta = 0.05$. In Figures 4(a) and 5(a), we plot the squared gradient norm v.s. the number of samples using 40 MDP trajectories. The upper and lower envelopes of the curves correspond to the 95 and 5 percentiles of the 40 curves. Similarly, we also plot the estimated variance

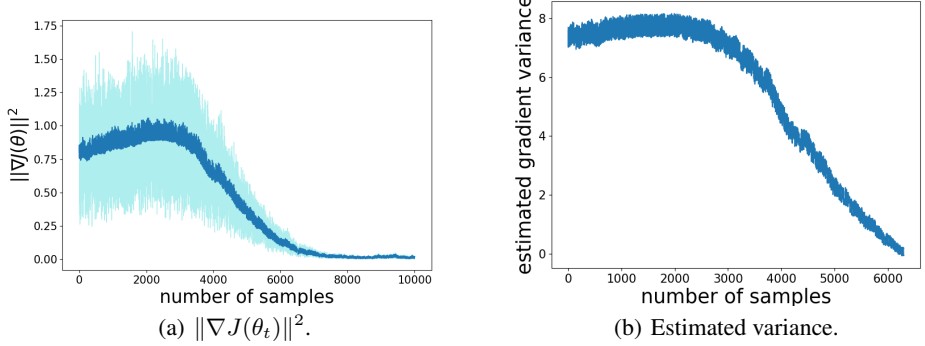

(a) $\|\nabla J(\theta_t)\|^2$.        (b) Estimated variance.

Figure 2: Garnet problem 2: $\mathcal{G}(3, 2, 3, 23)$.

$\|G_{t+1}(\theta_t, \omega_t) - \nabla J(\theta_t)\|^2$ of the stochastic update along the iterations using 50 samples at each time step. More specifically, we first run the algorithm to get a sequence of $\theta_t$ and $\omega_t$. Then we generate 50 different trajectories $O^i = (O_1^i, O_2^i, ..., O_t^i, ...)$ where $i = 1, ..., 50$, and use them to estimate the variance $\|G_{t+1}^i(\theta_t, \omega_t) - \nabla J(\theta_t)\|^2$ and plot $\frac{\sum_{i=1}^{50} \|G_{t+1}^i(\theta_t, \omega_t) - \nabla J(\theta_t)\|^2}{50}$ at each time $t$.

It can be seen that in both experiments, the gradient norm $\|\nabla J(\theta_t)\|$ converges to 0, i.e., the algorithm converges to a stationary point. The estimated variance also decreases to zero.

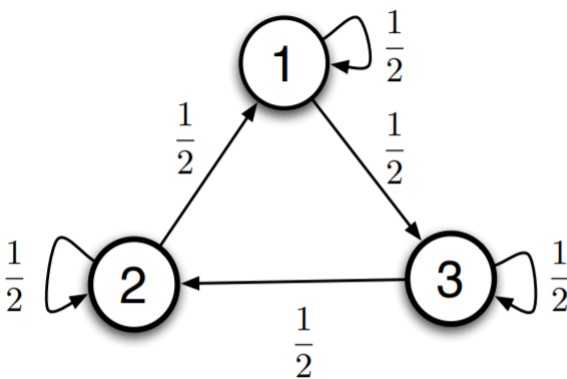

Figure 3: Spiral counter example.

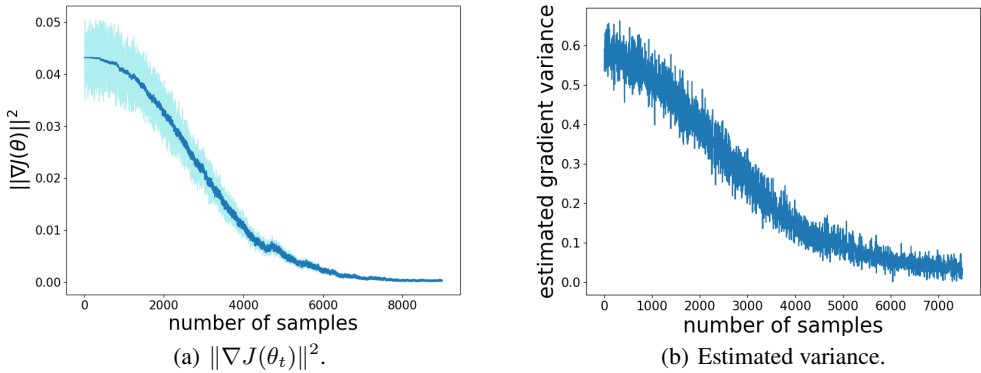

(a) $\|\nabla J(\theta_t)\|^2$.

(b) Estimated variance.

Figure 4: Spiral counter example 1:
$a = [0.94, -0.43, 0.18], b = [0.21, -0.52, 0.76]$.

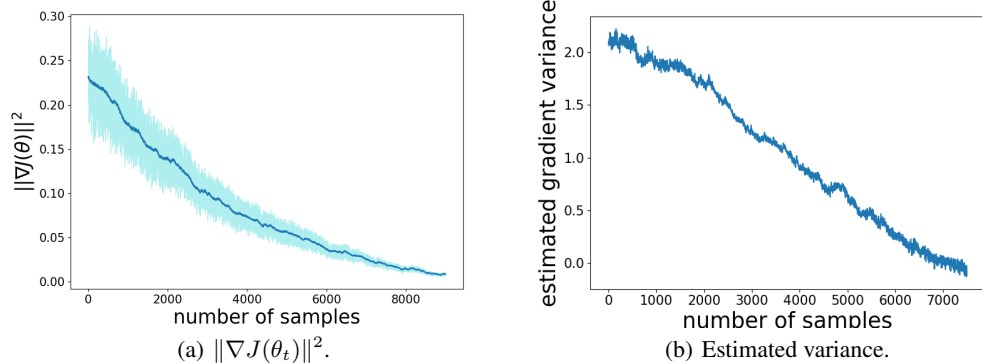

Figure 5: Spiral counter example 2:
$a = [0.21, -0.33, 0.29], b = [0.68, 0.41, 0.82].$