# OpenReview forum: "Non-Asymptotic Analysis for Two Time-scale TDC with General Smooth Function Approximation"
_NeurIPS.cc/2021/Conference — NeurIPS 2021 Poster_

### Official Review · Reviewer_bpa2 · 2021-07-02

**Rating:** 6
**Confidence:** 4

**Summary:**

This paper provides finite-sample analysis on the two time-scale TDC algorithm with non-linear function approximation, proposed by [5] Bhatnagar et al. Both the i.i.d. setting and the Markovian setting are considered. The convergence rate is near-optimal given the target MSPBE function is generally non-convex.

**Limitations And Societal Impact:**

The authors pointed out the limitation that the target function is generally non-convex and therefore this paper can only guarantee convergence toward a saddle point rather than a global minimum. There are no specific concerns with respect to its social impact.

**Main Review:**

The paper is clearly written and easy to understand.

Based on the asymptotic results given in [5] Bhatnagar et al., it is natural to consider the finite-sample analysis and the non-asymptotic rate $1 / \sqrt{T}$ achieved here is a bit surprisingly good. The downside though is that the whole analysis falls under the framework where the outside loop (the slower component $\theta$) is to optimize a general non-convex function with stochastic gradients, while the inner loop (the faster component $\omega$) is a linear stochastic approximation task (which is effectively strongly convex). Therefore the analysis can always be decomposed to first bounding the tracking error of the faster component $\omega$ to the optimal solution $\omega(\theta)$ while the slower component only suffers a small loss from the tracking error aside from the classical non-convex SGD analysis. The Markovian setting is great but also suffers from a similar novelty issue since there has been plenty of previous works dealing with the Markov noise under the uniform ergodicity assumption.

One suggestion is that the arguments in Line 539 and Line 594 that requires $\alpha$ to be much smaller than $\beta$ need to be stated more clearly, because there are too many constants involved there and it is better to write down an assumption on the ratio $\alpha / \beta$.

It would help highlight the technical contribution if the author can also compare the two time-scale techniques used here with those in some previous works:
1. (this presents the non-convex outer loop + convex inner loop framework) A Finite-Time Analysis of Two Time-Scale Actor-Critic Methods. NIPS'20
2. (a more general formulation) A Two-Timescale Stochastic Algorithm Framework for BilevelOptimization:  Complexity Analysis and Application to Actor-Critic. Arxiv 2007.05170



**Time Spent Reviewing:**

4

---

> ### Author Response · Authors · 2021-08-06
> **order-level better rate than existing literatures, which matches with the rate O(1/sqrt{T}) for a general smooth non-convex problem**
>
> We thank the reviewer for providing valuable feedback that helps us improve the quality of this paper. Below is a point-to-point response to the questions and comments raised by the reviewer.
>
> 1. Q: The analysis falls under the framework for a class of problems.
>
>     A: We agree with the reviewer that our analysis can be decomposed to first bounding the tracking error and then analyzing the convergence of a general non-convex function. We would like to highlight that our major technical novelty lies in a tighter analysis of the tracking error, and we provided an order-level better error bound. Specifically, the two references pointed out by the reviewer did not achieve $\mathcal O(1/\sqrt{T})$ or $\epsilon^{-2}$ as we did in our paper. Specifically, the complexity in [Wu et al. 2020] and [Hong et al. 2020] is $\epsilon^{-2.5}$, whereas ours is $\epsilon^{-2}$. The key idea of our tracking error analysis is to bound the tracking error in terms of the gradient norm $\nabla J(\theta)$, whereas in these papers, the bound on the tracking error is typically independent of the slow time-scale variable, which is not as tight. Our approach may also be useful to tighten the bounds in the suggested papers.
>
> 2. Q: The treatment of Markovian noise is not very novel.
>
>     A: We agree that using the uniform ergodicity and mixing argument to analyze the Markovian noise is not new in this paper. The Markovian setting is mainly included for completeness of the work, and our major novelty lies in analysis for two time-scale non-linear updates and non-convex objective, and a tight tracking error analysis, which can be seen in our proof for the i.i.d. setting.
>
> 3. Q: Provide the requirements on the ratio $\alpha/\beta$?
>
>     A: We will make it clear how $\alpha$ shall scale compared to $\beta$ in the main paper.
>
> 4. Q: Can you compare the technique with two additional reference papers?
>
>     A: We thank the reviewer for providing these related works. We will discuss these references in the paper.
>
>       Our problem is related to these two references, however, it is also different in several aspects. First of all, we provide a much tighter analysis, which achieves a complexity of $\epsilon^{-2}$, and is tighter than the complexity of $\epsilon^{-2.5}$ in [Wu et al. 2020] and [Hong et al. 2020]. Second, for the actor critic algorithm, the outer loop is policy optimization. The value function, although being nonconvex in the policy, enjoys linear convergence rate under the deterministic case, e.g., [Agarwal et al 2019] [Bhandari and Russo 2020], which suggests a much better geometry of the value function. Here, our objective function is the projected Bellman error, which does not necessarily have such a nice geometry.  Even though, for the general framework of non-convex outer loop + convex inner loop, our analysis provides a much tighter rate.

---

### Official Review · Reviewer_zbSi · 2021-07-16

**Rating:** 8
**Confidence:** 5

**Summary:**

This paper develops the non-asymptotic error bound for an important RL algorithm TDC with general non-linear function approximation. The TDC algorithm is a commonly used one for off-policy policy evaluation in practice. Existing studies on non-asymptotic error bound mostly focus on the case with linear function approximation. This work extends the study to the case with non-linear function approximation. Several new challenges are addressed in this paper, the nonconvexity of the objective function, non-linear two time-scale update rule, time-varying projection and a tight tracking error analysis. Both the iid and Markovian settings are investigated. The convergence rate provided is $\mathcal O(1/\sqrt(T))$.


**Limitations And Societal Impact:**

Some comments/questions are also listed below.

1. The experiments could be moved to the main body of the paper as there is still room.

2. In remark 2, it is discussed that the ratio $\alpha/\beta$ does not necessarily need to go to zero (in this paper alpha and beta are chosen to be constant at the order of $O(1/\sqrt(T)$). This is kind of counterintuitive. Some more elaboration on this issue could help.
In assumption 4, it is assumed that the MDP is uniformly ergodic. It is kind of a standard assumption in RL analysis when there is Markovian noise. It would be helpful to discuss what kind of MDPs satisfy this assumption.

3. Figs 4 and 5 in the appendix shall be aligned.

4. Page 25, it is assumed that the step size $\beta$ satisfies some condition $\beta\tau_\beta C_\phi^2\leq 1/4$, which should also be stated in the theorem.

**Main Review:**

Pros:
1. The proof generalizes the analysis for nonconvex optimization in [Ghadimi & Lan 2013] to the  TDC algorithm with a two time-scale structure, where the tracking error due to the two time-scale updates needs to be bounded, which is also their major technical novelty in the analysis. They bound the tracking error using the gradient norm $E[\|\nabla J(\theta_t)\|^2]$, which should also converge to zero. Other analysis of two time-scale algorithms either use a constant bound (resulting in a loose bound in the end) or using a mini-batch method (resulting in a large batch size ($1/\epsilon$). Their idea of bounding the tracking error using $E[\|\nabla J(\theta_t)\|^2]$, which is the quantity that is being bounded, is novel and interesting. This technique could be useful for developing non-asymptotic error bound of other two time-scale algorithms with a non-convex objective function.

2. Another novelty lies in the way they handle dependency between the projection tangent plane and the sample trajectory. For most existing studies with linear function approximation, the projection is fixed, and there does not exist such an issue. They first characterized the geometry of those functions, and then decoupled the dependency by exploiting the smoothness of the function approximator.

Cons:
There is no significant difference between the iid setting and the Markovian setting. The approach of handling the Markovian noise using a mixing argument is kind of standard in the current RL literature.

Quality: The complexity of $\mathcal O(1/\sqrt(T))$ looks natural. This result matches with the complexity of optimizing a general smooth non-convex function in [Ghadimi & Lan 2013]. The proof looks correct.

Clarity: This paper is well-written. I enjoyed reading this paper. The proof of sketch part is clear and informative, where the major steps and technical novelty are highlighted. The table of contents in the appendix is helpful to navigate in the appendix.

Significance: As discussed above, the idea of bounding the tracking error using the gradient norm is novel, and is useful for analyzing other two time-scale algorithms with non-convex objectives.





**Time Spent Reviewing:**

3

---

> ### Author Response · Authors · 2021-08-06
> **Response**
>
> We thank the reviewer for providing valuable feedback that helps us improve the quality of this paper. Below is a point-to-point response to the questions and comments raised by the reviewer.
>
> 1. Q: The treatment of Markovian noise is standard.
>
>     A: We agree that using the uniform ergodicity and mixing argument to analyze the Markovian noise is not new in this paper. Our major technical contributions lie in analyzing two time-scale non-linear updates and non-convex objective, and a tight analysis of the tracking error, which are presented in the i.i.d. setting. Extending the results to the Markovian setting is mainly for the completeness of the work.
>
> 2. Q: The experiment part can be moved to the main paper.
>
>     A: We will move the experimental results to the main paper.
>
> 3. Q: More elaboration on the ratio of step sizes.
>
>     A: We would like to emphasize that $\alpha/\beta\rightarrow 0$ is not necessarily required for two time-scale update rules to converge, e.g., [14] and [16] where linear two time-scale stochastic approximation was studied.  As shown in [14], the fast and slow time-scale updates can be decomposed asymptotically via a linear transformation so that the slow time-scale convergence rate is independent of the fast-time scale convergence rate. Though the linear transformation cannot be constructed for the non-linear problem here, we proved that the convergence rate is $\sqrt{T}$, which is the same as the rate of optimizing a smooth non-convex problem as in [12].
>
> 4. Q: What kind of MDP satisfies assumption 4?
>
>     A: The assumption on the MDP is a commonly used one. A Markov chain is uniformly ergodic if it is irreducible (i.e., it can arrive at any state from any state) and aperiodic ([Levin & Peres 2017]). Suppose there exists a policy that can map any state to any state with nonzero probability (i.e., irreducible) and get back to the same state aperiodically. Then as long as the behavior policy can take any action at any state with non-zero probability, the induced Markov chain remains irreducible and aperiodic, and is hence uniformly ergodic. We will make this more clear in our paper.
>
> 5. Q: A requirement of $\beta$ is missing in the main theorem statement.
>
>     A: The condition that $\beta\tau_\beta C^2_\phi\leq 1/4$ can be satisfied for a large $T$. We will make this clear in the theorem.

---

### Official Review · Reviewer_SQEZ · 2021-07-16

**Rating:** 6
**Confidence:** 2

**Summary:**

This paper analyses the convergence of two time-scale TDC with general smooth function approximation. Previous analysis mainly focuses the linear approximation, but the nonlinear approximation is more powerful and its convergence analysis is more challenging; thus, this work is theoretically solid. The analysis covers both i.i.d. settings and Markov settings.



**Limitations And Societal Impact:**

No negative societal impact

**Main Review:**

This paper analyses the convergence of two time-scale TDC with general smooth function approximation. Previous analysis mainly focuses the linear approximation, but the nonlinear approximation is more powerful and its convergence analysis is more challenging; thus, this work is theoretically solid. The analysis covers both i.i.d. settings and Markov settings.

No empirical study is carried out. If some reinforcement learning tasks are conducted to verify the theory, the results can be more convincing.

1.	It is mentioned that the projection step in Line 6 of algorithm 1 is for the convenience of the analysis. So, does removing the projection destroy the convergence or yield a worse convergence rate?
Typo: Line 103, Page 3, resent.


**Time Spent Reviewing:**

7

---

> ### Author Response · Authors · 2021-08-06
> **Response**
>
> We thank the reviewer for providing valuable feedback that helps us improve the quality of this paper. Below is a point-to-point response to the questions and comments raised by the reviewer.
>
> 1. Q: There are no experiments.
>
>     A: In this paper, our main focus is on the theoretical analysis of the TDC algorithm with non-linear function approximation in [5], and therefore, we put our numerical results in Appendix D. We will move our numerical results to the main paper. Some more numerical results can also be found in  [5].
>
> 2. Q: Is the projection step removable?
>
>     A: In our analysis, we need the projection to guarantee that $\omega$ is finite so that the gradient estimate $G_t(\theta,\omega)$ is Lipschitz in $\omega$. This projection step makes our tracking error analysis clean and clear without introducing tedious details on handling the boundedness of $\omega$, and makes it easier for readers to understand our major technical contribution in a careful and tight analysis of the tracking error and two time-scale non-linear update rule. Also, the projection radius was explicitly given in the paper for practitioners. Furthermore, the projection technique in practice usually leads to an improved stability of the algorithm.
>
>    The algorithm still converges asymptotically if the projection is removed, as was shown in [5], where TDC with non-linear function approximation was originally proposed. Without the projection step, we believe the same convergence rate of $\mathcal O(1/\sqrt{T})$ can still be obtained using the approach in [28], which however will complicate the proof.
>
> 3. Q: Typos.
>
>     A: We thank the reviewer for pointing out the typo, and we will correct it.

---

### Decision · Program_Chairs · 2021-09-27

**Decision:**

Accept (Poster)

**Comment:**

This paper provides non-asymptotic analysis for TDC algorithm with general smooth function approximation, which is technically sound and novel. The authors’ responses have well addressed the reviewers’ concerns. All the reviewers have reached a consensus on the acceptance of this work. We suggest that the authors further polish their paper to prepare a camera-ready version based on the reviewers’ comments.